



# Extreme Sea Levels in the Baltic Sea under Climate Change Scenarios. Part 1: Model Validation and Sensitivity

Christian Dieterich[1], Matthias Gröger[1], Lars Arneborg[1], and Helén C. Andersson[1]

[1]Swedish Meteorological and Hydrological Institute, Folkborgsvägen 17, 601 76 Norrköping, Sweden

**Correspondence:** Christian Dieterich (christian.dieterich@smhi.se)

**Abstract.** An ensemble of regional climate change scenarios for the Baltic Sea is validated and analyzed with respect to extreme sea levels (ESLs) in the recent past. The ERA40 reanalysis and five Coupled Model Intercomparison Project Phase 5 (CMIP5) global general circulation models (GCMs) have been downscaled with the coupled atmosphere-ice-ocean model RCA4-NEMO. Validation of 100-year return levels against observational estimates along the Swedish coast shows that the
model estimates are within the 95% confidence limits for most stations, except those on the west coast. The ensemble mean 100-year return levels turns out to be the best estimator with biases of less than 10 cm. The ensemble spread includes the 100-year return levels based on observations. A series of sensitivity studies explores how the choice of different parameterizations, open boundary conditions and atmospheric forcing affects the estimates of 100-year return levels. A small ensemble of different regional climate models (RCMs) forced with ERA40 shows the highest uncertainty in ESLs in the southwestern Baltic Sea and
in the northeastern part of the Bothnian Bay. Also the Skagerrak, Gulf of Finland and Gulf of Riga are sensitive to the choice of the RCM. A second ensemble of one RCM forced with different GCMs uncovers a lower sensitivity of ESLs against the variance introduced by different GCMs. The uncertainty in the estimates of 100-year return levels introduced by GCMs ranges from 20 cm to 40 cm at different stations. It is of similar size as the 95% confidence limits of 100-year return levels from observational records.

## 1 Introduction

The coastal area of the Baltic Sea is home to around 15 million people. Sea level rise (SLR) and sea level extremes in the densely populated areas are an immediate concern to the public, to authorities and to other stakeholders. In Sweden many thousand people live in areas that are at risk to be flooded during extreme storm surges (Perbeck, 2018). Fredriksson et al. (2017) have
shown that the exposure to storm surges has increased and if an event like Backafloden happened today the impact would be greater than back in 1872. In the past centuries changes in sea levels in the Baltic Sea have forced communities to build new harbors or move settlements (Ekman, 2009). Sea level had fallen relative to the land due to the glacial isostatic adjustment (GIA). That has lead to harbours falling dry with economic impact on local societies. Today's estimates for the land uplift





relative to the geoid range between -0.2 mm/a for the German and Polish coasts up to 9 mm/a at Höga Kusten (Jivall et al., 2016; Ågren and Svensson, 2011). The land is still rising since the last ice age.

Since the beginning of industrialization the global warming trend has caused an accelerating global mean sea level (GMSL) rise (Church et al., 2013). They give an average of 3.2 mm/a GMSL rise for the period 1993 to 2009. The main contribution

to GMSL rise has been from the expansion of the warming water in the global oceans. Melt water from glaciers and ice sheets that increase the amount of water in the global ocean has contributed another one third to the GMSL rise. To assess possible trajectories of climate change and related GMSL rise the CMIP5 project (Taylor et al., 2012) has coordinated an ensemble of model runs with GCMs. These models take into account, apart from natural forcing, representative concentration pathways (RCPs) of how much extra warming is projected at the end of the 21$^{st}$ century (van Vuuren et al., 2011). This ensemble of

global climate scenarios is extensively discussed in the fifth assessment report (AR5) of the IPCC (Stocker et al., 2013). The GMSL rise in the year 2100 relative to the period 1986 to 2005 ranges from 44 cm (RCP2.6) to 74 cm (RCP8.5), according to Church et al. (2013). The uncertainty for those estimates across the RCPs ranges from 28 cm (RCP2.6) to 98 cm (RCP8.5).

At the end of the century, relative to the period 1986 to 2005 the combined effects of GIA and GMSL rise are of the same order of magnitude. The median estimate of GMSL rise in the RCP8.5 scenario is canceled out on a line that divides the

Bothnian Sea from the Baltic Proper. North of it the GIA is dominating and sea level relative to land is falling. In the Gulf of Finland, Gulf of Riga, Baltic Proper, Arkona Basin, Danish Straits, Kattegat and Skagerrak sea level is projected to rise relative to land. For other RCPs with less anthropogenic warming the zero line of the combined effect would shift south-eastwards. With the most recent estimates including high end and extreme scenarios (e.g. Sweet et al., 2017) GMSL rise could also reach 250 cm in the year 2100 in which case there will be sea level rise relative to land all around the Baltic Sea.

Another factor that determines the mean sea level (MSL) of the Baltic Sea is related to the large-scale atmospheric circulation over the North Atlantic. Kauker and Meier (2003) have found a good correlation of the zonal wind component with the sea level at station Landsort. The sea level at Landsort is a good measure for the volume of water (or the averaged mean sea level) in the Baltic Sea (Matthäus and Franck, 1992). For the interannual variations Andersson (2002) has shown that sea level variation in Stockholm correlate significantly with the NAO index. For positive phases of the NAO, that are characterized by a more

zonal and a stronger atmospheric circulation the MSL of the Baltic Sea is expected to rise. According to AR5 (Stocker et al., 2013) the NAO is likely to become slightly more positive under projected climate change. That would translate to a possible rise of the MSL of the Baltic Sea. Recently, Karabil et al. (2018) found good correlation of interannual and decadal sea level variability in the Baltic Sea with the BANOS index that reflects more closely the variability in geostrophic wind in the entrance region of the Baltic Sea.

It has long been known (Ekman, 2009) that the sea level in the Baltic Sea is highest during winter. Samuelsson and Stigebrandt (1996) have shown that on the seasonal and shorter time scales sea level variations in the Baltic Sea are caused by large-scale atmospheric circulation patterns. Together with a potential increase in positive NAO phases and a concurrent increase in the strength of low pressure systems (Schneidereit et al., 2007; Pinto et al., 2009) higher ESLs in the Baltic Sea during winter must be anticipated. However, Meier (2006) has found that ESL may rise faster than MSL even without significant changes in the

wind field in downscaled projections of the Baltic Sea.





Analyses of ESLs by Weisse et al. (2014) at specific locations along the European coast, including the Baltic Sea, have shown an increase in the past 100 years. Their projections show a continuing increase of ESLs with MSL rise being the main contributor. They expect decadal variability to contribute to ESL changes in the near future. In their study Vousdoukas et al. (2016) have projected ESLs for the entire coastline of Europe using the bias corrected output of a shallow water model driven

with an ensemble of eight CMIP5 models and two RCP scenarios. Both Vousdoukas et al. (2016) and Wahl et al. (2017) discuss the uncertainty of ESLs introduced by the method used to estimate the sea level with long return periods. Wahl et al. (2017) also set into relation the uncertainty of the methodology to the uncertainty introduced by SLR scenarios and conclude that specially for the near future the uncertainty from the choice of the method is dominating. While Wahl et al. (2017) present a global analysis Eelsalu et al. (2014) has shown for the Estonian coast that no method for extreme value estimation was capable

to accommodate all observed and hindcast extremes and that the spread among different methods can be substantial.

A number of modeling studies have focused on ESLs in the Baltic Sea. Meier et al. (2004) downscaled two SRES scenarios (Special Report on Emission Scenarios) with two different GCMs. They found large uncertainties in ESLs both from the use of different GCMs and the use of different SLR scenarios. Kowalewski and Kowalewska-Kalkowska (2017) showed that in general, modeled sea level variability in the Baltic Sea can be improved by an increase in resolution. Gräwe and Burchard

(2012) used a high resolution model for the western Baltic Sea and could show that the increased resolution ($\sim 1\,\mathrm{km}$) allowed the realistic simulation of extremes in the Danish Straits. They also could show that MSL rise causes a nonlinear response in sea level extremes by up to O($10\,\mathrm{cm}$) in shallow and narrow locations in the western and southern Baltic Sea. Hieronymus et al. (2017) investigated the contribution of various forcing mechanisms on the sea level in the North Sea and Baltic Sea and showed that contributions from local wind forcing, atmospheric pressure, as well as remote sea level forcing are important for the Baltic

sea levels, and that they interact in a non-linear way to increase the variability. They also showed that the influence of external sea level forcing on periods less than 50 days is damped inside the Baltic Sea.

One advantage of regional models versus global models is the higher resolution that can be used to resolve orography and bathymetry. The atmospheric and oceanic dynamics that interact with the regional features give rise to the specific characteristics of the region (e.g. Stein and Alpert, 1993; Feser et al., 2011; Jeworrek et al., 2017). To faithfully model sea level

dynamics in the Baltic Sea, Kattegat and Skagerrak a reasonable representation of the driving agents wind and pressure is a minimum requirement. The atmosphere component RCA4 has been shown to yield a good climate compared to observational data sets (Kjellström et al., 2016; Strandberg et al., 2014). Wind from a A1B scenario downscaled with RCA4-NEMO has been analyzed and compared to other RCM results by Ganske et al. (2016). They found low wind speeds in RCA4-NEMO for the highest 99 percentile for the North Sea compared to other RCMs. Dieterich et al. (2013) have shown that the mean wind speed

in RCA4-NEMO compares well with observations. Gröger et al. (2015) compared wind speed from RCA4-NEMO with corresponding values from an uncoupled run with RCA4. The largest improvements in wind speed in the coupled model were found in the winter season in regions where the Baltic Sea is covered with sea ice. In uncoupled RCA4 runs the SST is determined by the ocean component of global hindcast simulations that only coarsely resolves the Baltic Sea. This points to an added value using a coupled model for modeling sea level in the Baltic Sea. That is specially true for ESLs that are caused by storms,





predominantly in winter time (Samuelsson and Stigebrandt, 1996) when air-sea interaction is underrepresented (Gröger et al., 2015).

This study aims at an analysis of ESLs for the Baltic Sea, Kattegat and Skagerrak based on model simulations for the past and future climates. The model used in this study is a regional atmosphere-ice-ocean model that was used to downscale model solutions of the CMIP5 ensemble. The model is briefly described in Sect. 2. A validation with emphasize on ESLs is presented in Sect. 3. Sections 4 discusses the sensitivity of the ESLs against changes in sub-grid scale parameterizations, open boundary conditions and atmospheric forcing. An attempt is made in Sect. 5 to map the uncertainty due to RCMs and the one introduced by GCMs. A final Sect. 6 discusses the results and identifies topics that need to addressed for future progress in modeling ESLs in the Baltic Sea. Conclusions can be found at the end.

Projections of ESLs for the 21[st] century and their sources of uncertainty based on the same ensemble are discussed in a companion paper.

## 2 Model Description

A coupled RCM was used to investigate mean and ESLs and how they might change under scenario assumptions. The RCM is set up for the North Sea and Baltic Sea region and consists of an atmosphere component and an ocean component. The atmosphere model covers the whole of Europe as defined by CORDEX and is used at the SMHI in different resolutions. The one used in the coupled version has a resolution of 0.22° and 40 levels. The ocean component includes an ice model and resolves the North Sea and the Baltic Sea with 2 nautical miles and with 56 vertical levels. This coupled system is called RCA4-NEMO and has been introduced by Wang et al. (2015). The version used here for the scenario simulations is the one evaluated by Dieterich et al. (2013, 2019) without the the river routing model.

Other aspects of this model ensemble have been discussed previously: major baltic inflows (Schimanke et al., 2014), air-sea coupling (Gröger et al., 2015), changes in wind speed and direction (Ganske et al., 2016), snow-bands (Jeworrek et al., 2017), model intercomparison (Pätsch et al., 2017), changes in heat fluxes (Dieterich et al., 2019).

A regional model comes at the expense of having to formulate boundary conditions that allow information from the global atmosphere and the global ocean to enter the model domain. The treatment of the open boundaries follows the strategies laid out in Wang et al. (2015); Dieterich et al. (2019). The sea surface height (SSH) along the open boundaries of the ocean component determines the averaged SSH in the regional model domain. Together with the atmospheric forcing, the SSH information on the open boundaries also contributes to the sea surface variability on time-scales from hours (Büchmann et al., 2011) to decades (Karabil et al., 2017).

To represent the tides in the regional model 11 harmonic constituents from the global tidal model at the Oregon State University (Egbert et al., 2010) are applied as open boundary conditions.

For sensitivity runs discussed in Sect. 4 the hourly SSH from a storm surge model covering the North East Atlantic is added to the other components of the SSH on the open boundary.





The monthly SSH prescribed along the open boundaries is derived from the global solutions of the OGCMs and transfers the information of seasonal, interannual and decadal SSH variability from the global to the regional scale. Details of the procedure are described in Dieterich et al. (2019). The varying SSH of the OGCMs in the northern North Sea represents characteristics of the regional circulation. A high SSH along the European shelf might indicate a weakening North Atlantic Current in the global

model (e.g. Saenko et al., 2017). This leads to different averaged SSHs in the regional model which in turn might interact with sea level dynamics on a more local scale (e.g. Gräwe and Burchard, 2012; Pelling et al., 2013). The water level in the Kattegat and the Danish Straits also has consequences for the ventilation and the ecosystem of the Baltic Sea (e.g. Hordoir et al., 2015; Arneborg, 2016; Meier et al., 2016).

In order to obtain an ensemble of sea level solutions for the present climate, we have downscaled the historical periods of

a number of CMIP5 GCMs from 1961 to 2005 in addition to the ERA40 reanalysis. The scenario part (2006 and onward) of these CMIP5 runs represent different representative concentration pathways (RCPs). They have been downscaled too and the ESLs in the Baltic Sea in these projections are discussed in a companion paper.

**Table 1.** Ensemble of regional climate experiments for the North Sea and Baltic Sea region. The table lists the ERA40 reanalysis and the historical periods of five CMIP5 GMCs that have been downscaled with RCA4-NEMO.

| experiment | historical | comments |
|---|---|---|
| RCA4-NEMO ERA40 | 1961 - 2009 | standard experiment |
| RCA4-NEMO MPI-ESM-LR | 1961 - 2005 | |
| RCA4-NEMO EC-EARTH | 1961 - 2005 | |
| RCA4-NEMO GFDL-ESM2M | 1961 - 2005 | |
| RCA4-NEMO HadGEM2-ES | 1961 - 2005 | |
| RCA4-NEMO IPSL-CM5A-MR | 1961 - 2005 | |

The RCA4-NEMO runs discussed in the next sections are summarized in Table 1. This small ensemble offers a first insight into the uncertainty that is generated due to different large scale conditions, represented by the GCMs.

To set into relation the uncertainty that is inherent in the RCA4-NEMO ensemble forced with different GCMs a second group of experiments is analyzed that uses one GCM but different RCMs. These experiments are listed in Table 2. The RCMs are not independent of each other, but originate from different model setups that are used at the SMHI. The first five setups, except RCA4-NEMO-alt use the same ocean component NEMO-Nordic. RCA4-NEMO-alt differs from the standard experiment RCA4-NEMO ERA40 by using a different ocean component. Some of the relevant differences are lateral mixing

along geopotential surfaces, instead of isopycnic ones. Also the alternative NEMO-Nordic uses mixing coefficients according to Smagorinsky (1963). The bottom friction is larger and lateral walls impose a free-slip condition. The model setups RCA4-NEMO-1hr and RCA4-NEMO-50km differ from RCA4-NEMO by an 1-hourly coupling and a 0.44° resolution RCA4, respectively. For more details on NEMO-Nordic 3.6 see Hordoir et al. (2019) and Höglund et al. (2017) for the model setup used in the STORMWINDS project.





**Table 2.** Sensitivity experiments with different RCMs forced with ERA40. RCA4-NEMO-1hr ERA40 is the same setup as RCA4-NEMO ERA40 but the atmosphere- and ice-ocean-components are coupled every hour. RCA4-NEMO-alt is another coupled setup, where the ocean component is replaced by an alternative NEMO-Nordic setup. RCA4-NEMO-50km is a setup where the RCA4-NEMO is run at a resolution of 0.44°. NEMO-Nordic indicates the ocean component used in the regular RCA4-NEMO setup. Here it is used as an ocean-only setup that has been forced with the output of RCA4 ERA40. NEMO-Nordic 3.6 ERA40 is the ocean-only setup validated by Hordoir et al. (2019). STORMWINDS ERA40 is an ocean-only setup for the Baltic Sea that has been used in the STORMWINDS project (Höglund et al., 2017).

| experiment | historical | comments |
|---|---|---|
| RCA4-NEMO ERA40 | 1961 - 2009 | standard experiment (Table 1) |
| RCA4-NEMO-1hr ERA40 | 1961 - 2009 | standard with 1-hourly coupling |
| RCA4-NEMO-alt ERA40 | 1961 - 2009 | standard with alternative NEMO-Nordic |
| RCA4-NEMO-50km ERA40 | 1961 - 2009 | standard with RCA4 0.44° resolution |
| NEMO-Nordic ERA40 | 1961 - 2009 | standard ocean-only experiment |
| NEMO-Nordic 3.6 ERA40 | 1961 - 2005 | NEMO-Nordic 3.6 (Hordoir et al., 2019) |
| STORMWINDS ERA40 | 1961 - 2005 | NEMO-Nordic 3.6 (Höglund et al., 2017) |

**Table 3.** Sensitivity experiments with different atmospheric forcing for different RCMs. RCA4-NEMO-1hr ERA40 is the same setup as RCA4-NEMO ERA40 but the atmosphere- and ice-ocean-components are coupled every hour. NEMO-Nordic ERA40 and NEMO-Nordic ERA-interim are two ocean-only setups forced with the output of RCA4 ERA40 and RCA4 ERA-interim, respectively. NEMO-Nordic interpolated is the same setup as NEMO-Nordic ERA40 but with linearly interpolated forcing (see text for more details). NEMO-Nordic 3.6 ERA40 and NEMO-Nordic 3.6 EURO4M (Hordoir et al., 2019) are two the ocean-only setups forced with ERA40 and EURO4M, respectively.

| experiment | historical | comments |
|---|---|---|
| NEMO-Nordic ERA40 | 1961 - 2009 | standard ocean-only experiment (Table 2) |
| NEMO-Nordic ERA-interim | 1979 - 2011 | standard forced with ERA-interim |
| NEMO-Nordic interpolated | 1961 - 2009 | standard with linearly interpolated forcing |
| NEMO-Nordic 3.6 ERA40 | 1961 - 2005 | NEMO-Nordic 3.6 (Table 2) |
| NEMO-Nordic 3.6 EURO4M | 1961 - 2005 | NEMO-Nordic 3.6 forced with EURO4M |
| RCA4-NEMO ERA40 | 1961 - 2009 | standard experiment (Table 1) |
| RCA4-NEMO-1hr ERA40 | 1961 - 2009 | standard with 1-hourly coupling (Table 2) |





Different RCMs have been forced with different reanalysis datasets and Table 3 gives an overview of these sensitivity experiments. Differences between the different RCMs may be larger than differences between different atmospheric forcing datasets. That should be kept in mind when interpreting the results. NEMO-Nordic ERA-interim uses the ERA-interim reanalysis instead of the ERA40 reanalysis. Usually the atmospheric forcing like wind, pressure and so on are available every three hours. Either

these fields are kept constant or they are linearly interpolated. In a sensitivity run NEMO-Nordic interpolated the forcing fields are linearly interpolated in time and the ocean model experiences a smooth change in the forcing variables. The model setup NEMO-Nordic 3.6 has been driven with two different atmospheric reanalyses. The EURO4M reanalysis uses an atmosphere model with a higher resolution and has been shown to improve on results of the ERA40 reanalysis (Dahlgren et al., 2016). The reference setup RCA4-NEMO ERA40 may be compared with a setup RCA4-NEMO-1hr ERA40 where the atmosphere and

ice-ocean components exchange fluxes and surface temperatures every hour. This sensitivity study may answer the question whether a more frequent coupling than 3-hourly is necessary.

**Table 4.** Sensitivity runs forced with RCA4 ERA40. The runs differ from each other by the open boundary conditions that have been applied at the open boundaries of NEMO-Nordic. All runs in this table are ocean-only experiments. These sensitivity runs have been described in more detail in Dieterich et al. (2019).

| experiment | historical | open boundary conditions for NEMO-Nordic |
|---|---|---|
| NEMO-Nordic interpolated | 1961 - 2009 | standard (Table 3) |
| NEMO-Nordic ORAS4 | 1961 - 2009 | standard with monthly T, S, SSH from ORAS4 |
| NEMO-Nordic ORAS4 b | 1961 - 2009 | standard with monthly T, S, SSH, transports from ORAS4 |
| NEMO-Nordic ORAS4 c | 1979 - 2009 | standard with ORAS4 b with storm surge model |
| NEMO-Nordic Surge | 1979 - 2009 | standard with monthly T, S. SSH from storm surge model |
| NEMO-Nordic Surge b | 1979 - 2009 | standard with Surge, SSH years randomly rearranged |
| NEMO-Nordic NOT | 1961 - 2009 | standard without tides |

A group of sensitivity experiments that explores the influence of the open boundary conditions on ESLs is summarized in Table 4. The experiments ORAS4, ORAS4 b and ORAS4 c resolve the conditions in the northern North Sea with a monthly resolution. That introduces low frequency variability in the open boundaries of the ocean model, like the NAO. The experiments

ORAS4 c, Surge and Surge b add hourly SSH from a storm surge model to the ocean model. This technique adds high frequency SSH variability that has been generated in the North East Atlantic and traveled to the open boundary of the ocean model. The experiment NEMO-Nordic NOT is set up to check the influence of tidal forcing on ESLs in the Baltic Sea.

Table 5 lists sensitivity studies where miscellaneous model parameters and different parameterizations have been varied to estimate their influence on ESLs in the Baltic Sea. NEMO-Nordic viscous employs constant lateral viscosity with a har-

monic operator, while NEMO-Nordic no-slip uses a viscosity coefficients that are large where the velocity field shows a large shear (Smagorinsky, 1963). The experiment NEMO-Nordic free-slip can be compared to NEMO-Nordic no-slip. Constant lateral viscosity and no-slip conditions are the standard for all experiments. These sensitivity experiments are intended to identify





**Table 5.** Sensitivity runs that use different model parameters, parameterizations, mean sea level and river discharge. All runs in this table are ocean-only experiments. The sensitivity run NEMO-Nordic viscous uses constant lateral viscosity, while NEMO-Nordic no-slip uses coefficients according to Smagorinsky (1963). NEMO-Nordic free-slip differs by the slip conditions along lateral walls compared to NEMO-Nordic no-slip. In NEMO-Nordic MSL the MSL is 58 cm higher compared to NEMO-Nordic interpolated (cf. Table 3). NEMO-Nordic E-HYPE and NEMO-Nordic discharge differ by the river discharge, which is $O(1500 \, \text{m}^3/\text{a})$ higher in NEMO-Nordic E-HYPE and which is used in all other experiments.

| experiment | historical | open boundary conditions for NEMO-Nordic |
|---|---|---|
| NEMO-Nordic viscous | 1961 - 2009 | harmonic viscosity |
| NEMO-Nordic no-slip | 1961 - 2009 | no-slip conditions |
| NEMO-Nordic free-slip | 1961 - 2009 | free-slip conditions |
| NEMO-Nordic interpolated | 1961 - 2009 | interpolated forcing |
| NEMO-Nordic MSL | 1961 - 2009 | interpolated with MSL + 58 cm |
| NEMO-Nordic E-HYPE | 1961 - 2009 | interpolated with E-HYPE based discharge |
| NEMO-Nordic discharge | 1961 - 2009 | interpolated with less discharge |

the influence of the lateral slip conditions on ESLs. The next two experiments NEMO-Nordic interpolated and NEMO-Nordic MSL answer whether the MSL has an impact on ESLs. How do ESLs react on variations in the river discharge can be learned from the last two experiments NEMO-Nordic E-HYPE and NEMO-Nordic discharge. The former model run uses the standard discharge as any other experiment, while experiment NEMO-Nordic discharge uses the river discharge that was used by Meier et al. (2004).

## 3 Model Validation

### 3.1 Mean Sea Levels

The five different GCMs used for the regional downscaling exhibit different MSLs, where the regional model domain has it's open boundaries. For this reason, the MSL averaged over the regional model domain varies between -20 cm to 160 cm among different ensemble members. To match observed MSLs in the Baltic Sea the model results need to be adjusted, using observed time series and estimates of their MSL. Here we have used the estimated MSL at station Landsort for the year 1986 as a reference. It has been determined with a linear regression using long-term observations (Hammarklint, 2009). The model results for sea level are each corrected with a constant, so that the modeled MSL at Landsort in the period 1970 to 1999 matches the estimated MSL from observations.

The modeled and observed mean sea surface for the period 1970 to 1999 is compared in Fig. 1. The increase of the mean sea surface from the Skagerrak to the Kattegat through the Baltic Proper into the Gulf of Finland and to the Bothnian Bay is clearly reproduced. The differences from observational estimates are within 5 cm (Table 6) for both the hindcast simulation and the





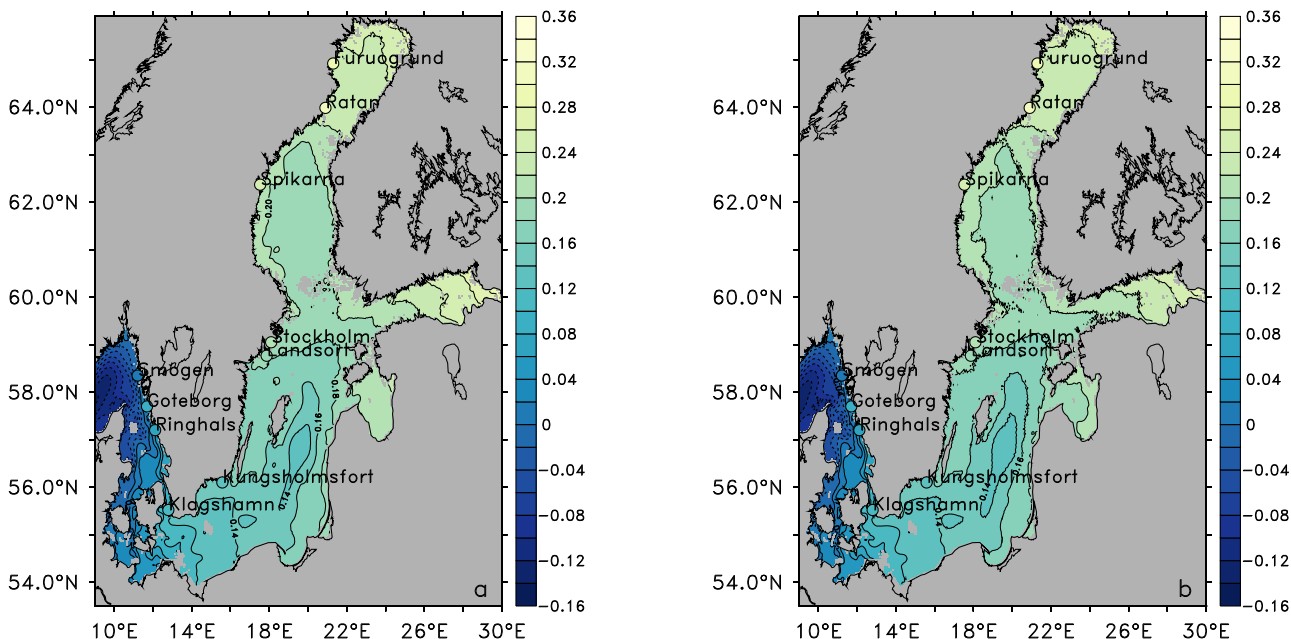

**Figure 1.** Mean sea level [cm] for the period 1970 to 1999 for the hindcast RCA4-NEMO ERA40 (a) and the ensemble mean (b). The colored dots indicate MSL estimated from long-term observations (WISKI, 2017).

ensemble mean of the historical period, except for station Ratan. There the model underestimates the mean sea surface by 7 cm. Meier et al. (2004) have shown a good correspondence of their model results with MSL estimates by Ekman and Mäkinen (1996). Compared to our model ensemble Meier et al. (2004) apply observed SSH in the Kattegat as boundary condition to their model. We speculate that this is the main reason for the somewhat larger discrepancy between our model results and

observational estimates.

ESLs are measured against the mean sea surface and it is therefore important to get the mean sea surface right. A confirmation of a representative ensemble for sea level estimates is the fact that the mean sea surface of the ensemble mean differs only slightly from the one found in the hindcast simulation. The maximum differences between hindcast and ensemble mean are seen at station Spikarna with less than 1 cm.

Since the five different GCMs plus the ERA40 reanalysis provide a range of distinct atmospheric conditions it is unlikely that the atmospheric forcing is responsible for the biases seen in the modeled mean sea surface of the Baltic Sea. On the other hand Meier et al. (2004) have shown in a sensitivity study, that an increase of 30% in wind speed does increase the MSL gradient along the Swedish coast from Smögen to Furuögrund by around 4 cm. That would bring the three northernmost stations in Table 6 closer to the estimated mean sea surface based on the WISKI network. This could indicate that the model

system used here generally produces low wind speeds at least when inferred from the mean sea surface of the Baltic Sea.

In a second sensitivity study Meier et al. (2004) increased the river discharge to the Baltic Sea by 34%. The pattern is different from the one caused by an increased wind speed and would fit with the data presented in Fig. 1 and Table 6 because





**Table 6.** Mean sea levels [cm] in the period 1970 to 1999 for selected stations along the Swedish coast (cf. Fig. 1). The values in brackets indicate model biases relative to the observational estimates. Estimates from the observational network WISKI (2017) are based on a long-term regression at that station. Model results are the MSL at that station referenced to station Landsort.

| station | WISKI | RCA4-NEMO ERA40 | RCA4-NEMO ensemble mean |
|---|---|---|---|
| Furuögrund | 27.9 | 23.9 [-4.0] | 24.0 [-3.9] |
| Ratan | 30.0 | 22.9 [-7.1] | 23.0 [-7.0] |
| Spikarna | 26.6 | 20.9 [-5.7] | 21.6 [-5.0] |
| Stockholm | 22.1 | 18.7 [-3.4] | 18.9 [-3.2] |
| Landsort | 18.8 | 18.5 [-0.3] | 18.5 [-0.3] |
| Kungsholmsfort | 13.7 | 15.4 [1.7] | 15.5 [1.8] |
| Klagshamn | 9.6 | 10.8 [1.2] | 11.0 [1.4] |
| Ringhals | 6.5 | 4.8 [-1.7] | 5.2 [-1.3] |
| Göteborg | 6.3 | 3.5 [-2.8] | 3.4 [-2.9] |
| Smögen | 1.1 | 1.8 [0.7] | 0.5 [-0.6] |

the Baltic Proper would show a mean sea surface around 1 cm higher than the Kattegat and the Bothnian Bay. Our modeled mean sea surface with positive biases in the Baltic Proper suggests that the model system has a fresh bias, which has been found by Dieterich et al. (2013, 2019) as well.

## 3.2 Extreme Sea Levels

An investigation on ESLs along the Swedish coast by Södling and Nerheim (2017) has shown that ESLs in the region are best estimated using a generalized extreme value (GEV) distribution with the blockmaxima method. Using the same technique the different model runs have been analyzed to produce return sea levels for different return periods at the sea level stations operated by the SMHI (WISKI database). These are listed in Table 7 together with the values estimated from observations by Södling and Nerheim (2017).

The table contains the estimates from the ERA40 hindcast with the coupled model RCA4-NEMO in lines two. In the northern Baltic Sea the agreement between RCA4-NEMO ERA40 results and the observational estimates is good. In the central Baltic Sea the model underestimates return levels. In the southern Baltic Sea the agreement is good. On the west coast of Sweden the largest differences are found between model and observational estimates.

Lines three in Table 7 show return levels estimated from an ocean-only model that has been driven with a downscaled ERA40

reanalysis. Except for the west coast the model setup NEMO-Nordic 3.6 ERA40 produces lower return levels than the coupled model RCA4-NEMO ERA40.

Fig. 2 shows a comparison of ESLs from seven different model configurations (Table 2) with corresponding estimates from the WISKI database (Södling and Nerheim, 2017). Generally, the model estimates of the 100-year return levels are lower than





**Table 7.** Extreme sea levels [cm] using a GEV distribution with the blockmaxima method. First line according to Södling and Nerheim (2017, Table 5.3), second line from RCA4-NEMO ERA40 (1961 to 2005), third line from NEMO-Nordic 3.6 ERA40 (1961 to 2005). The values in brackets indicate the 95% confidence interval.

| return period | 10 years | 100 years | 200 years |
|---|---|---|---|
| Furuögrund | 113 [103 to 122] | 145 [122 to 165] | 153 [125 to 176] |
| RCA4-NEMO ERA40 | 113 [103 to 123] | 144 [116 to 172] | 152 [116 to 188] |
| NEMO-Nordic 3.6 ERA40 | 92 [84 to 100] | 113 [93 to 134] | 118 [93 to 144] |
| Ratan | 104 [97 to 111] | 130 [113 to 143] | 136 [116 to 152] |
| RCA4-NEMO ERA40 | 105 [96 to 114] | 131 [110 to 152] | 138 [111 to 164] |
| NEMO-Nordic 3.6 ERA40 | 87 [80 to 94] | 106 [89 to 122] | 110 [89 to 130] |
| Spikarna | 94 [86 to 101] | 120 [103 to 134] | 126 [105 to 143] |
| RCA4-NEMO ERA40 | 84 [79 to 89] | 94 [88 to 99] | 95 [89 to 101] |
| NEMO-Nordic 3.6 ERA40 | 78 [71 to 86] | 94 [79 to 109] | 97 [79 to 115] |
| Stockholm | 81 [75 to 86] | 102 [88 to 113] | 107 [90 to 120] |
| RCA4-NEMO ERA40 | 73 [68 to 78] | 85 [75 to 95] | 87 [76 to 99] |
| NEMO-Nordic 3.6 ERA40 | 64 [59 to 69] | 77 [69 to 88] | 81 [68 to 93] |
| Landsort | 72 [67 to 77] | 92 [80 to 104] | 98 [81 to 111] |
| RCA4-NEMO ERA40 | 71 [66 to 76] | 83 [73 to 93] | 85 [74 to 97] |
| NEMO-Nordic 3.6 ERA40 | 62 [57 to 67] | 75 [65 to 85] | 78 [65 to 90] |
| Kungsholmsfort | 96 [90 to 103] | 120 [122 to 142] | 126 [108 to 140] |
| RCA4-NEMO ERA40 | 95 [87 to 103] | 116 [99 to 134] | 121 [99 to 141] |
| NEMO-Nordic 3.6 ERA40 | 77 [70 to 83] | 95 [78 to 111] | 99 [78 to 120] |
| Klagshamn | 116 [108 to 123] | 135 [122 to 142] | 138 [123 to 144] |
| RCA4-NEMO ERA40 | 103 [93 to 112] | 130 [102 to 158] | 137 [101 to 173] |
| NEMO-Nordic 3.6 ERA40 | 83 [76 to 90] | 102 [87 to 117] | 106 [88 to 125] |
| Ringhals/Varberg | 122 [114 to 129] | 149 [132 to 163] | 155 [134 to 171] |
| RCA4-NEMO ERA40 | 83 [75 to 90] | 102 [80 to 124] | 107 [79 to 136] |
| NEMO-Nordic 3.6 ERA40 | 102 [93 to 111] | 120 [105 to 136] | 124 [106 to 143] |
| Smögen | 122 [115 to 128] | 142 [128 to 152] | 147 [130 to 158] |
| RCA4-NEMO ERA40 | 105 [100 to 111] | 120 [103 to 137] | 124 [102 to 146] |
| NEMO-Nordic 3.6 ERA40 | 105 [98 to 112] | 125 [104 to 146] | 130 [103 to 158] |





those from the observations. The same is true for the 20-year return levels (not shown). In the Bothnian Bay (Furuögrund and Ratan) and in the southern Baltic Sea (Kungsholmsfort and Klagshamn) the model estimates from the coupled model RCA4-NEMO ERA40 are within the confidence limits of the observational estimates. In the central Baltic Sea (Stockholm and Landsort) all models estimate somewhat lower return levels than the observations would suggest. On the west coast (Ringhals

and Smögen) none of the model setups is reproducing the 100-year return levels. Since some model configurations include a storm surge model in the formulation of the open boundary conditions the cause for the underestimation of ESLs lies probably either in the atmospheric forcing or the unresolved effects in the ocean, due to an insufficient (2 nautical miles) resolution. At those stations where the coupled model matches the estimates derived from observations the ESLs are to a large degree determined by atmospheric forcing. The different treatment of the air-sea interaction between a coupled atmosphere-ocean

model and an ocean-only model would explain most of the discrepancy. Among the coupled model runs the one with a reduced atmospheric resolution is clearly an outlier. It shows that all along the Swedish coast good atmospheric information is essential to estimate ESLs. The exception is station Smögen, where the storm surges are generated further west under open ocean conditions and an atmospheric resolution of 50 km produces a surge of roughly the same height as a resolution of 25 km.

## 4 Model Sensitivity

The light gray shading in Fig. 2 is the uncertainty generated by different RCMs. Comparing it with the colored shading shows that overall, the RCMs disagree more than the confidence limit of the GEV estimation. This is true however only for the mixed ensemble of coupled and uncoupled RCMs. Clearly, the two groups are clustered and the ensemble is not normally distributed. The uncertainty within the first three coupled RCMs and the three uncoupled RCMs in Table 2 and Fig. 2 is much smaller.

ESLs in the Baltic Sea are sensitive to details of how physics and dynamics are implemented in the numerical model. It is

well known that bottom friction has a major impact on the amplitude and phase of sea level variations (Gräwe and Burchard, 2012). In this section a series of sensitivity runs are presented that are set up to explore how ESLs depend on different aspects of model implementation and forcing.

### 4.1 Decadal Variability

From the relatively small spread (O(20 cm)) among the estimates in Fig. 3 it can be concluded that the ESLs are not very

sensitive against the choice of different long-term (30 years or longer) analysis intervals. This might be true however only for return periods shorter than 100 years. The estimates differ most between the two mutually exclusive 25-year periods for the first and second half of the 50 year model run. In the Bothnian Bay and in the Kattegat the extremes are higher in the first half. In the Baltic Proper the extremes are higher in the second half. The same tendency although smaller is seen between other periods that cover the first and second half of the historical period. The case of the two 25-year periods can be interpreted as

the point where the length of the analysis period became too short to yield a robust estimate of a 100-year return level. ESLs in the northern Baltic Sea seem to be most affected by different choices of analysis periods. That might have to do with the ice cover, which is known to be sensitive to decadal variability (Jevrejeva et al., 2003). During positive phases of NAO the



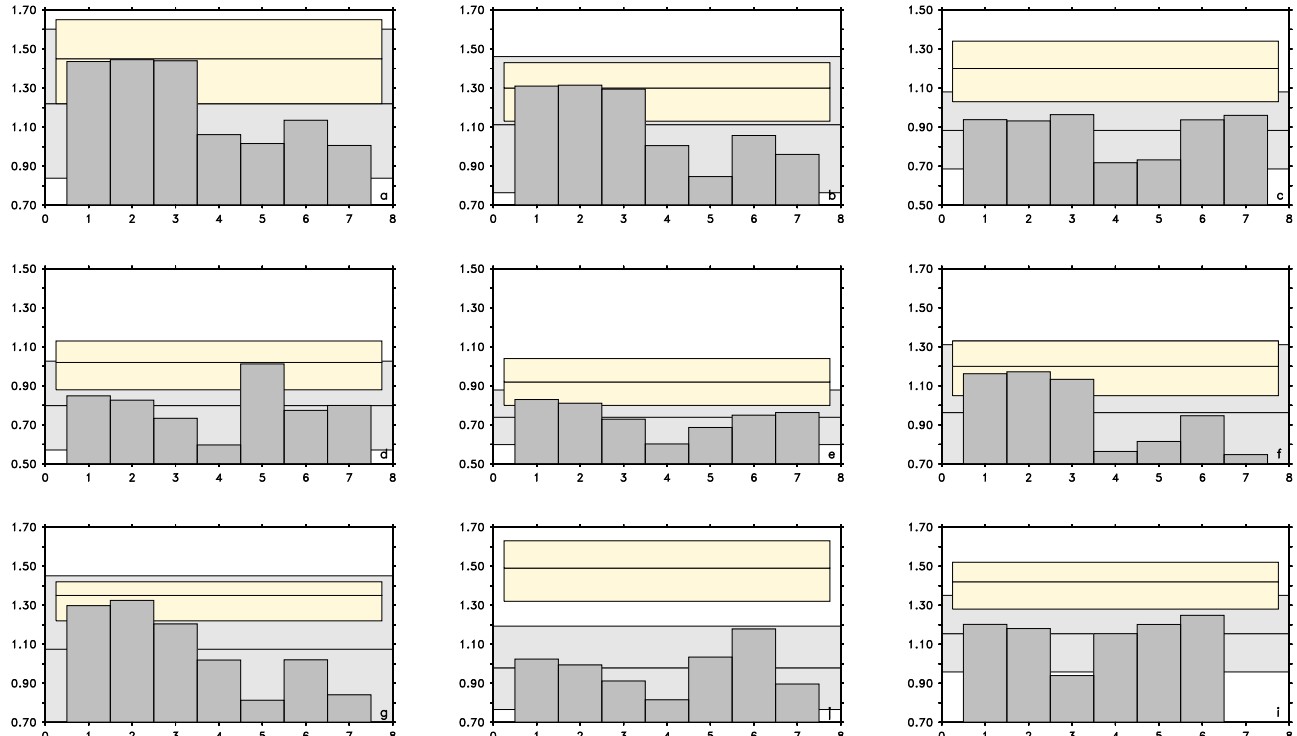

**Figure 2.** Extreme sea levels with 100-year return period for nine stations along the Swedish coast: Furuögrund (a), Ratan (b), Spikarna (c), Stockholm (d), Landsort (e), Kungsholmsfort (f), Klagshamn (g), Ringhals (j), Smögen (i). The different realizations have been estimated using different model setups: RCA4-NEMO ERA40 (1), RCA4-NEMO-1h (2), RCA4-NEMO-alt (3), RCA4-NEMO-50km (4), NEMO-Nordic (5), NEMO-Nordic 3.6 (6), STORMWINDS (7). The different experiments are summarized in Table 2. The horizontal lines represent the mean of the different estimates. The lightly shaded area around it shows 1.96 times the ensemble dispersion. The estimates from the WISKI database by Södling and Nerheim (2017) with the 95% confidence intervals are drawn as colored areas.

Baltic Sea tends to experience mild winters with less ice cover (Omstedt and Chen, 2001) together with stronger westerlies. This situation promotes the momentum transfer from the atmosphere to the ocean and the generation of storm surges.

## 4.2 Atmospheric Forcing

The influence of atmospheric forcing on ESLs is shown in Fig. 4. ESLs of seven different model runs are compared. However,
5    there are different models involved as well that behave quite differently (cf. Fig. 2). What can be deduced from Fig. 4 is the sensitivity of ESLs at station Smögen depending on whether the ERA40 or the ERA-interim reanalysis has been used to force the ocean model. Other differences, even between different model setups have a minor ($< O(10\,\mathrm{cm})$) impact. At station Furuögrund ESLs turn out differently depending on whether the ERA40 or the EURO4M reanalysis has been used as forcing. Together with the EURO4M experiment the two coupled runs show similar 100-year return levels, which are all much more
10    realistic then those from the other experiments. In the Bothnian Bay at least a part of this difference needs to be attributed





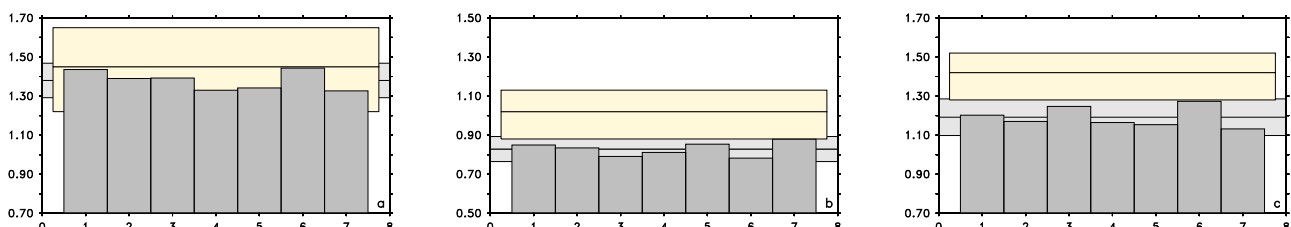

**Figure 3.** Extreme sea levels from RCA4-NEMO ERA40 with 100-year return period for three stations: Furuögrund (a), Stockholm (b), Smögen (c). The different analysis periods are 1961 to 2005 (1), 1970 to 2009 (2), 1961 to 1989 (3), 1970 to 1999 (4), 1980 to 2009 (5), 1961 to 1984 (6) and 1985 to 2009 (7). The horizontal line represents the mean of the different estimates and the lightly shaded area around it is the 95% confidence interval.

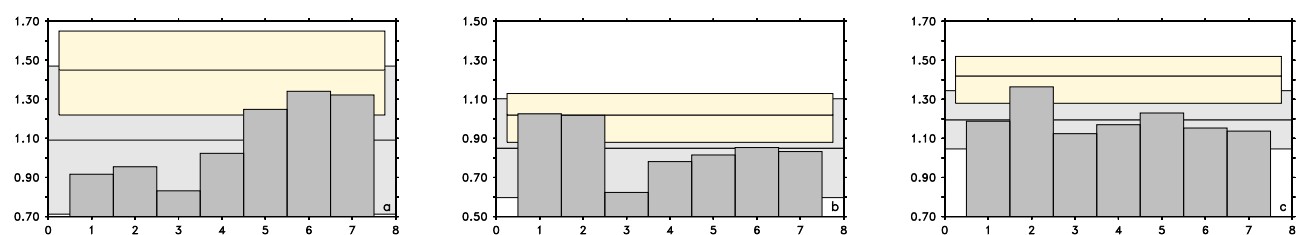

**Figure 4.** Extreme sea levels as in Fig. 3. The different sensitivity runs are NEMO-Nordic ERA40 (1), NEMO-Nordic ERA-interim (2), NEMO-Nordic interpolated (3), NEMO-Nordic 3.6 ERA40 (4), NEMO-Nordic 3.6 EURO4M (5), RCA4-NEMO ERA40 (6), RCA4-NEMO-1hr ERA40 (7). Table 3 gives an overview of the experiments. The common analysis period was 1980 to 2009, except for NEMO-Nordic 3.6 ERA40 it was 1980 to 2005.

to the higher wind speeds found in the EURO4M data set (Dahlgren et al., 2016) and the coupled experiments (Gröger et al., 2015). The experiment NEMO-Nordic interpolated confirms that with lower return levels compared to the regular experiment NEMO-Nordic ERA40, where the forcing was not interpolated between timesteps. In the latter case the stepwise forcing includes higher harmonics that generate high frequency gravity waves that add to the sea levels extremes. As can be seen from
5  Fig. 4 ESLs at Stockholm are O(40 cm) higher if high frequency noise is present in the forcing. On the other hand the 1-hourly coupling in RCA4-NEMO-1hr ERA40 compared to it's reference run RCA4-NEMO ERA40 does not generate different return levels. The hypothesis here could have been that a more frequent update of the atmospheric forcing would improve the sea level simulation. Apparently, it is sufficient for this model setup to update the path of the storms that generate sea level extremes every 3 hours. Other stations from the WISKI network do not show an improvement either (not shown).

10  **4.3  Open Boundary Conditions**

The need to formulate open boundary conditions at the boundary of the computational domain of the model introduces an additional model sensitivity. To look into the effects of what type of information is available at the open boundary of the model





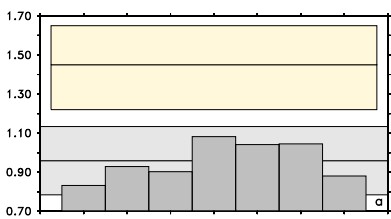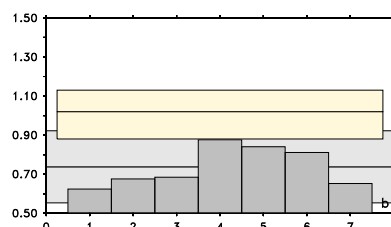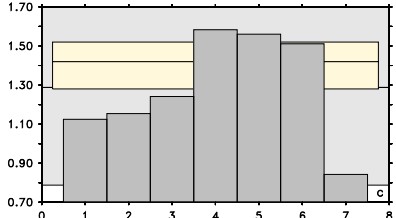

**Figure 5.** Extreme sea levels as in Fig. 3. The different bars refer to the experiments NEMO-Nordic interpolated (1), NEMO-Nordic ORAS4 (2), NEMO-Nordic ORAS4 b (3), NEMO-Nordic ORAS4 c (4), NEMO-Nordic Surge (5), NEMO-Nordic Surge b (6), NEMO-Nordic NOT (7). The different experiments are listed in Table 4. The common analysis period was 1980 to 2009.

domain a number of sensitivity runs (cf. Table 4) were performed. The regular open boundary that is used in the ensemble of scenarios discussed in Sect. 2 provides monthly mean temperature, salinity and sea surface height.

At all stations in Fig. 5 the effect of an additional storm surge model (experiments ORAS4 c, Surge, Surge b) is visible in the ESL estimates. Generally, ESLs are higher by O(20 cm) with the additional information on the barotropic answer of the

North East Atlantic to atmospheric disturbances. The largest effect is seen at station Smögen with 100-year return levels O(40 cm) higher than without storm surges generated and imported from the North East Atlantic. The difference between ESLs from the experiments ORAS4 c, Surge, Surge b is small. The presence of hourly sea level variability, even in the case it is out of phase with the atmospheric forcing (Surge b), provides nearly the same increase in ESLs as the deterministically driven model (Surge). On the contrary, tidal forcing on the open boundaries does not affect the 100-year return levels in the Baltic Sea. Only

at station Smögen is there a significant contribution of O(30 cm) to the extremes compared to the model setup NOT. The use of temporally resolved temperature, salinity and SSH on the open boundaries in experiments ORAS4 and ORAS4 b leads to somewhat higher 100-year return levels. In the Baltic Sea the effect is probably related to the NAO that is resolved in the open boundary conditions. At station Smögen an interaction of sea level dynamics with a higher recirculation in the Skagerrak in ORAS4 and specially in ORAS4 b might add to the difference of O(10 cm). What is interesting to note comparing experiments

ORAS4, ORAS4 c and Surge is the vanishing influence of the long time scales in the open boundary conditions (ORAS4 c and Surge) as soon as there is high frequency information provided on the open boundaries.

### 4.4 Model Parameters

The influence of the water depth, lateral friction and other model parameters on the 100-year return levels is shown in Fig. 6.

How the horizontal viscosity is represented in the model has an effect on ESLs. This can be deduced from the two experi-

ments NEMO-Nordic viscous and NEMO-Nordic no-slip. The former uses harmonic viscosity along the geopotential surfaces and the latter uses viscosity coefficients calculated according to Smagorinsky (1963). In the case NEMO-Nordic no-slip the selective viscosity prevents the degradation of gradients near the coast, where the ESLs are measured and leads to much higher O(10 cm) ESLs. At station Smögen the difference in the 100-year return levels is O(30 cm).





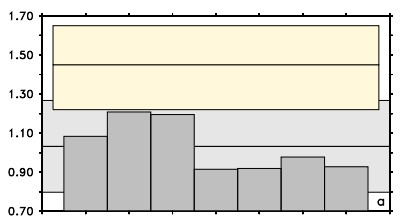 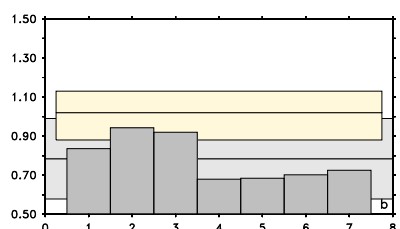 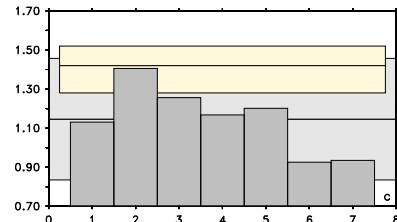

**Figure 6.** Extreme sea levels as in Fig. 3. The different sensitivity runs are NEMO-Nordic viscous (1), NEMO-Nordic no-slip (2), NEMO-Nordic free-slip (3), NEMO-Nordic interpolated (4), NEMO-Nordic MSL (5), NEMO-Nordic E-HYPE (6), NEMO-Nordic discharge (7). Table 5 gives an overview of the experiments. The common analysis period was 1961 to 2005.

Experiments NEMO-Nordic no-slip and NEMO-Nordic free-slip differ by the use of the slip conditions along the lateral walls. In the Baltic Sea the impact on ESLs is negligible. At station Smögen there is however a large effect of O(15 cm). Presumably the near shore ESLs interact with the near surface velocity field which is more distorted in the no-slip case and lead to higher ESLs. However, there is also the opposite effect in a different model setup (not shown). There, the no-slip

conditions lead to O(20 cm) lower ESLs in the Baltic Sea compared to the same model with free-slip conditions. This situation shows the difficulty and limitation of these sensitivity studies. The ESLs do not depend on different model parameters in an simple way. Depending on where in the parameter space resulting ESLs are compared can lead to opposite conclusions.

The influence of MSL on ESLs is shown in Fig. 6 for experiments four and five. The experiment NEMO-Nordic MSL has a higher MSL (58 cm) than NEMO-Nordic interpolated. At none of the stations shown in Fig. 6 is there a marked effect on ESLs

due to differing MSLs. This is in agreement with a study by Hieronymus et al. (2018) where the parameters of the GEV used to estimate ESLs do not change with MSL. A small effect can be seen at station Smögen where ESLs are higher with a lower MSL. This is in accordance with theory where shallower regions exhibit higher sea level signals (Pelling et al., 2013).

The last two experiments NEMO-Nordic E-HYPE and NEMO-Nordic discharge in Fig. 6 show the sensitivity of ESLs against the river discharge. NEMO-Nordic E-HYPE with a higher O(1500 m³/a) freshwater input than NEMO-Nordic dis-

charge shows a minor increase of ESLs in the Bothnian Bay. The freshwater signal is also visible in the MSL, which is 2 mm lower in the Bothnian Bay and 0.2 mm lower in the Baltic Proper. Since the spatial and temporal changes of the river discharge are different in the two experiments the effect of the higher freshwater input in NEMO-Nordic E-HYPE is masked by other processes.

As an overview Fig. 2 shows ESLs from seven different model configurations. Not all stations show the same sensitivity.

At stations Furuögrund and Ratan the coupled models with an atmospheric resolution of 0.22° shows much higher extremes than the uncoupled models. In this case all models were driven with the ERA40 hindcast. The coupled models translate the atmospheric momentum more efficiently into ESLs, compared to ocean-only models that employ a bulk formula to calculate wind stress. Additionally a different sea ice cover in the coupled and uncoupled models might explain differences in the Bothnian Bay. Also at stations Kungsholmsfort and Klagshamn the first three models show more realistic ESLs. Uncoupled

models generally produce too low ESLs and all models fail to reproduce ESLs as high as those estimated from observations on





the Swedish west coast. At station Smögen ESLs are much less sensitive to the choice of the model system, according to the sensitivity study presented in Fig. 2.

## 5   Model Uncertainty

In this section the GCM uncertainty is compared to uncertainty that comes with the RCM. With RCM uncertainty we mean the

range of solutions for ESLs that arise from the use of different model formulations or choice of parameters as shown in Fig. 2.

### 5.1   100-year Return Levels

In Fig. 7 the 100-year return levels for the five historical periods at nine sea levels stations along the Swedish coast are compared to the results from the downscaled hindcast RCA4-NEMO ERA40 and to observational estimates. One argument to use an ensemble of model runs is to gain insight into the spread of possible solutions. Additionally, for a meaningful ensemble

the ensemble mean should be better than individual model members. In this figure it becomes apparent that the ensemble mean is closer to the estimates from the WISKI database at all stations compared to individual model runs. Even though none of the historical model runs could have been compared to observations directly, Fig. 7 shows that the statistics of ESLs are realistic. As in Fig. 2 there are stations where the modeled 100-year return levels are underestimated. Except for stations Gothenburg and Smögen the ensemble mean of 100-year return levels are within the 95% confidence limits of the observational estimates.

It is therefore not wrong to assume that the ensemble averaged ESLs are from the same distribution as the ones estimated from observations.

The confidence limits for the 100-year return levels in Fig. 7 are based on how well the theoretical distributions can be approximated by the sampled ones. In the end it is a matter of how long the available timeseries are to estimate return levels. Long time series are rare and are usually available from a few stations only. So, natural variability tends to be underrepresented

in observed timeseries. Another way to produce a measure of uncertainty is the use of a model ensemble. It allows to estimate the spread or the dispersion of the ensemble for the whole model domain. There is still the need for reasonably long time series but the uncertainty in the ensemble can be reduced by increasing the number of ensemble members. The question is then how the spread in the ensemble is related to the confidence limits of the return levels estimated by the blockmaxima method with the GEV distribution.

Fig. 7 shows that 1.96 times the ensemble dispersion is larger than the 95% confidence limits for the GEV estimates for four stations. In the Baltic Proper and on the west coast the two measures are comparable. These are also the stations that are least sensitive to model formulation (Fig. 2) and atmospheric forcing (Fig. 4). At other stations the uncertainty in the ensemble could be reduced by increasing the number of ensemble members.

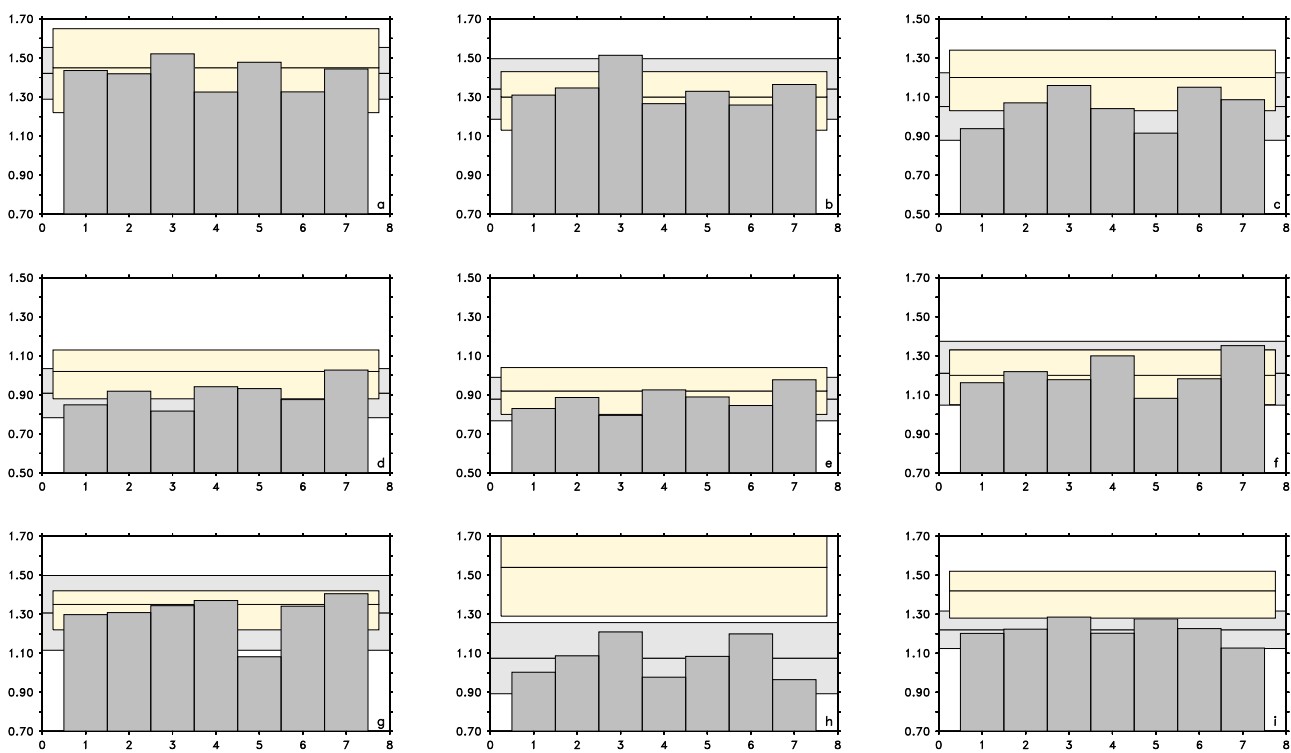

**Figure 7.** Extreme sea levels with 100-year return period for nine stations along the Swedish coast: Furuögrund (a), Ratan (b), Spikarna (c), Stockholm (d), Landsort (e), Kungsholmsfort (f), Klagshamn (g), Goteborg (h), Smögen (i). The different realizations have been estimated from RCA4-NEMO ERA40 (1), RCA4-NEMO ensemble mean HISTORICAL (2), RCA4-NEMO MPI-ESM-LR HISTORICAL (3), RCA4-NEMO EC-EARTH HISTORICAL (4), RCA4-NEMO GFDL-ESM2M HISTORICAL (5), RCA4-NEMO HadGEM2-ES HISTORICAL (6), RCA4-NEMO IPSL-CM5A-MR HISTORICAL (7) (cf. Table 1). The horizontal lines and shaded areas are as in Fig. 2. All model estimates are based on the common historical period 1961 to 2005.

## 5.2 The 99.9 Percentile Sea Levels

To be able to map the ESLs and their uncertainty for the whole Baltic Sea we turn to somewhat less extreme sea levels than the 100-year return levels. Figure 8 compares the warning levels used by the SMHI (Table 8) to the mean of the uppermost 99.9% of sea levels within the time period 1970 to 1999.

5    The individual estimates for the mean 99.9 percentile of ESLs (Fig. 8) agree very well with each other. 95% of all values are within no more than O(15 cm). This is the estimate of GCM uncertainty based on the 99.9 percentile. It is much lower than the 20 to 40 cm disagreement among the 100-year return levels in different GCMs (Fig. 7). The uncertainty estimates based on the 99.9 percentile are therefore minimum estimates for the uncertainty attributed to GCMs. Figure 8 indicates that the 99.9 percentile are close to warning level 1 used at the SMHI.





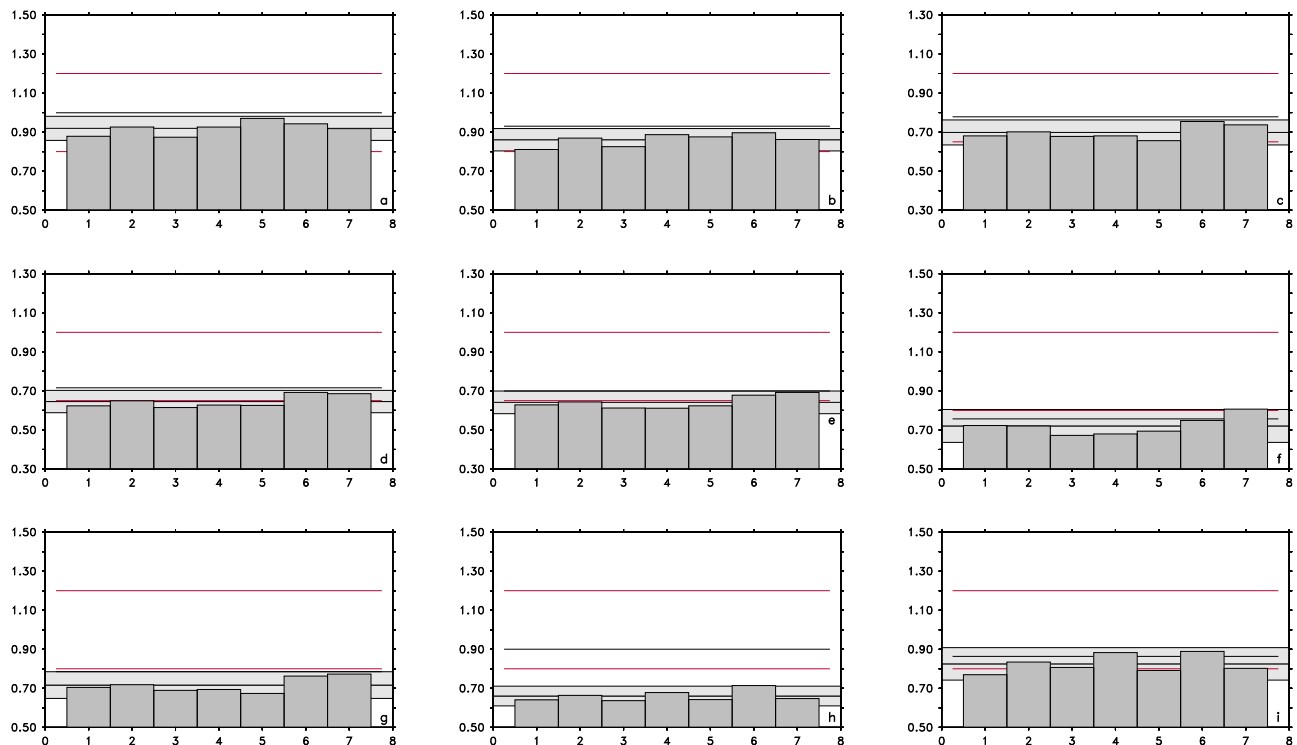

**Figure 8.** Mean 99.9 percentile of ESLs for nine stations along the Swedish coast: Furuögrund (a), Ratan (b), Spikarna (c), Stockholm (d), Landsort (e), Kungsholmsfort (f), Klagshamn (g), Goteborg (h), Smögen (i). The different realizations have been estimated from RCA4-NEMO ERA40 (1), RCA4-NEMO ensemble mean HISTORICAL (2), RCA4-NEMO MPI-ESM-LR HISTORICAL (3), RCA4-NEMO EC-EARTH HISTORICAL (4), RCA4-NEMO GFDL-ESM2M HISTORICAL (5), RCA4-NEMO HadGEM2-ES HISTORICAL (6), RCA4-NEMO IPSL-CM5A-MR HISTORICAL (7). The horizontal lines and shaded areas are as in Fig. 2. The red lines represent SMHI's warning level 1 and warning level 2, respectively. The horizontal black line is the 99.9 percentile of the observations. All estimates are based on the common historical period 1970 to 1999.

**Table 8.** Warning level for the Baltic Sea, Kattegat and Skagerrak used at the SMHI (Schöld et al., 2017). If sea level is predicted to be higher or equal to a specific warning level a public warning is issued. Warning levels are given relative to the mean sea level.

| region | warning level 1 | warning level 2 |
|---|---|---|
| West and South Coast | $\geq 80$ cm | $\geq 120$ cm |
| Bothnian Sea, Baltic Proper | $\geq 65$ cm | $\geq 100$ cm |
| Bothnian Bay | $\geq 80$ cm | $\geq 120$ cm |

The 99.9 percentile of ESLs are shown in Fig. 9. They are between 70 cm and 120 cm higher than the mean sea surface. The most extreme sea levels occur at the eastern end of the Gulf of Finland and the northern end of the Bothnian Bay. In the





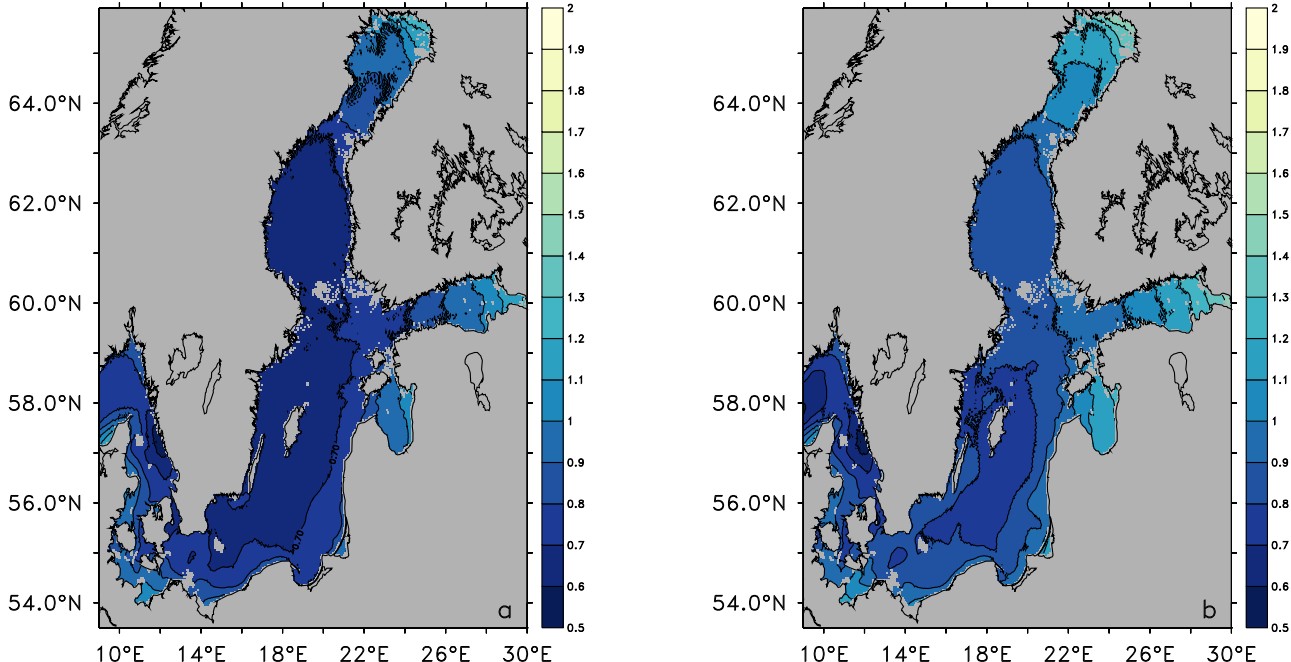

**Figure 9.** Ensemble median of the 99.9 percentile of ESLs [m] for the period 1970 to 1999 relative to the mean sea surface (a) and relative to bedrock (b).

western part of the Kattegat, the southwestern Baltic Sea and The Gulf of Riga the 99.9 percentile are up to 100 cm higher than the mean sea surface.

Since the mean sea surface is increasing towards the east and the north the 99.9 percentile relative to bedrock shows a more pronounced gradient towards the east and the north. Non-linear effects in the Bothnian Bay and the Gulf of Finland can be

identified by the crowding of contour lines.

While Fig. 9 shows the median of the GCM ensemble, Fig. 10 shows the likely range (5% to 95% range) of the GCM and the RCM ensemble. Both sources of uncertainty depicted in Fig. 10 yield the largest disagreement in ESLs in the northeastern part of the Bothnian Bay. Ensemble members driven with different GCMs show different sea ice covers that introduce an uncertainty in the transfer the horizontal momentum of the wind to the momentum of the barotropic motion in the ocean. The

10 RCMs among themselves also show a variety of patterns in ice cover. Additionally, the momentum transfer is implemented differently between coupled and uncoupled RCMs which adds to the uncertainty in ESLs in the northeastern Bothnian Bay. For the RCMs that uncertainty amounts to 50 cm. It can be reduced to 25 cm for the coupled-only RCM ensemble (not shown).

The shallower (< 100 m) parts of the Baltic Proper also are more uncertain in the model solutions than deeper parts. In shallow areas a small uncertainty in barotropic transport makes a relatively larger signal in the sea surface elevation than in

deeper water. Interestingly, the Bay of Riga and the Bays along the German and Polish coasts show high disagreement among different RCM solutions but not among GCMs. That points to the importance of the details in how atmospheric variability





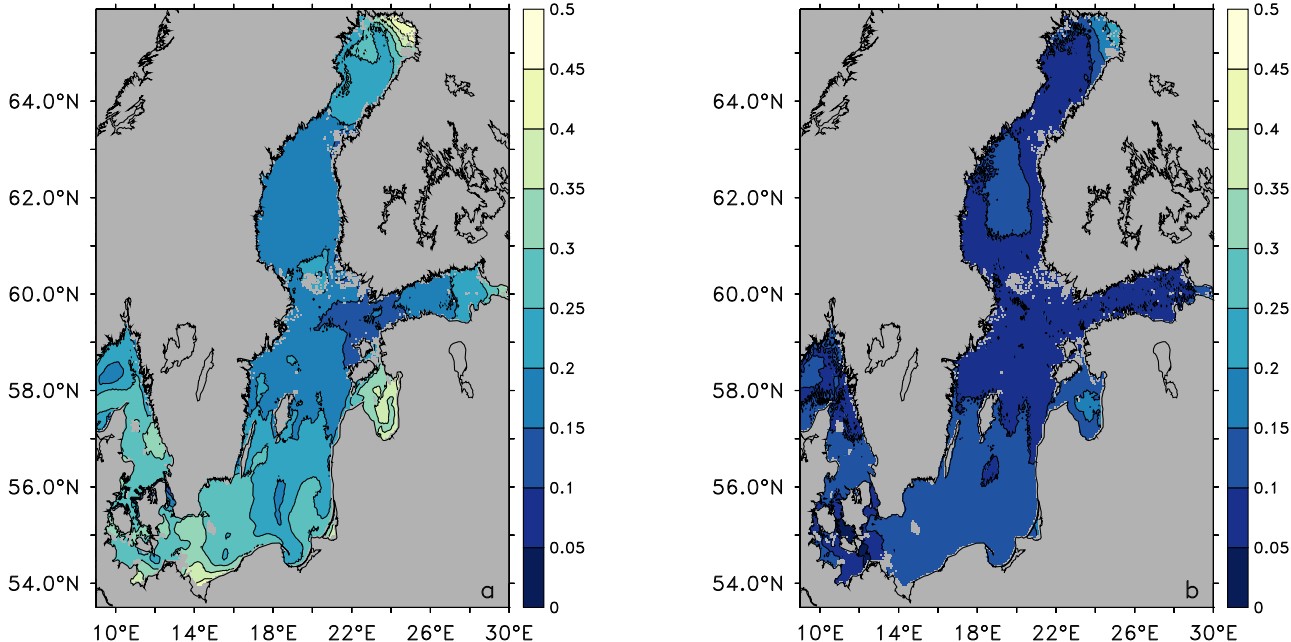

**Figure 10.** Uncertainty of ESL [m] for the period 1970 to 1999 for the RCM ensemble (a) and the GCM ensemble (b). The uncertainty is defined here as the likely range (5% to 95% range of ensemble solutions).

is translated into ESLs by the RCMs. The disagreement disappears for the coupled-only RCM ensemble. In both regions the disagreement, together with the extremes themselves, are higher closer to the coast. Since topography, resolution and forcing are the same in these regions the disagreement comes from a higher setup in shallower coastal regions, which is more sensitive to wind stress and bottom stress. Since the spatial scales are relatively small here nonlinear amplification of small differences

becomes increasingly important.

Generally for the Baltic Sea, Kattegat and Skagerrak the RCMs contribute about double the uncertainty compared to what the GCM uncertainty shows. There may be different explanations. From Fig. 2 it is obvious that the RCM ensemble shows a clustering between lower and higher extremes. Those solutions with the uncoupled RCMs for example (not shown) do not show any disagreement in the northeastern Bothnian Bay or in the eastern part of the Gulf of Finland. On the other hand it

shows a higher degree of uncertainty on the Swedish side of the Kattegat. The ensemble of RCM solutions is too small to provide a robust basis to estimate uncertainty. It should be understood as an upper bound for RCM uncertainty. Future model development should aim to reduce the disagreement.

## 6  Discussion

The validation of the modeled ESLs has shown that the ensemble mean of the historical period compares well to observational

estimates, except for the Swedish west coast (cf. Table 9). The modeled ESLs are lower than the ones inferred from obser-





vations. The results are however compatible with the assumption that the model generates the same distribution of ESLs as the observed ones. The biases in ESLs for the ensemble mean are smaller than O(10 cm), except for stations Gothenburg and Smögen.

**Table 9.** Extreme sea levels [cm] with 100-year return period (1961 to 2005) for selected stations along the Swedish coast (cf. Fig. 7). The values in brackets indicate model biases relative to the observational estimates. Note that the results in this table are from model runs that don't use a storm surge model.

| station | WISKI | RCA4-NEMO ERA40 | RCA4-NEMO ensemble mean |
|---|---|---|---|
| Furuögrund | 145 | 144 [-1] | 142 [-3] |
| Ratan | 130 | 131 [+1] | 135 [+5] |
| Spikarna | 120 | 94 [-26] | 107 [-13] |
| Stockholm | 102 | 85 [-17] | 92 [-10] |
| Landsort | 92 | 83 [-9] | 89 [-3] |
| Kungsholmsfort | 120 | 116 [-4] | 122 [+2] |
| Klagshamn | 135 | 130 [-5] | 131 [-4] |
| Göteborg | 154 | 100 [-54] | 109 [-45] |
| Smögen | 142 | 120 [-22] | 122 [-20] |

Table 9 also shows that the ensemble mean estimates are closer to the estimates based on observations than those calculated

from the single ERA40 hindcast. By means of sensitivity studies model deficiencies could be identified. At station Smögen the model is sensitive to sea level variability that is generated in the North East Atlantic. In the standard model configuration this information is not provided. The model results do improve however, when extra variability from the North Sea is present. At station Stockholm the model shows ESLs that are affected by processes and geography the model does not resolve properly. This can serve to formulate hypotheses for the development of improved model versions.

On the list of model improvements in the RCM is the reduction of the fresh bias in the Baltic Sea that would bring the mean sea surface closer to the observed one and thus reduce the underestimation of ESLs in the Kattegat and the northern Baltic Sea. Another model deficiency that needs to be addressed is the too low wind speed in the highest 99 percentile that are responsible for the generation of ESLs in the Baltic Sea.

A regional model can only to some extent represent effects that are caused by local characteristics of the bathymetry and

orography. Impact models are necessary to prolong the chain of climate downscaling and close the gap of spatial scales into the realm of ESLs in specific locations and climate change adaption on site. Johansson et al. (2017) have shown that the estimates of ESLs in Gothenburg are very sensitive to how far away the measurements are taken from the open sea. The resolution of the RCM is too coarse to resolve the river mouth of the Göta älv in Gothenburg where sea level measurements are taken. Some model development however can be envisioned for regional models that potentially improve ESL estimation. Drying and

wetting of adjacent low-lying land can help to more realistically represent the energy budget of storm surges. A wave model





could improve the representation how momentum is transferred from the atmosphere to the ocean and vice versa. Today, most coupled regional models treat the sea surface between the atmosphere and the ocean as an interface to exchange fluxes of momentum, energy and matter. A wave model can be integrated as an individual component in a coupled system to describe in more detail the air-sea exchange of momentum, energy and matter. A wave model would also help to integrate the Stokes

Drift into an ocean model that would allow interaction between the near-surface flow field and the storm surges (wave setup). Eelsalu et al. (2014) have shown that wave setup on ESLs are visible by the clustering of return levels along the Estonian coast. With a high resolution model for the German Bight Arns et al. (2017) have shown that the interaction between SLR, tides, storm surges and wind waves increase the 100-year return levels more than SLR alone would suggest.

In this study we presented a validation and analysis of simulated ESLs for the Baltic Sea, Kattegat and Skagerrak. To

calculate regional sea level scenarios we have downscaled five members of the CMIP5 ensemble for the historical period. A second part of the study discusses the sea level projections for the 21$^{st}$ century according to three different RCPs. The ensemble spread within the regional climate ensemble allows us to assess the uncertainty that is inherent to different GCM solutions. One source of uncertainty, which is missing from our ensemble is how different RCMs influence the uncertainty of ESLs. An approximate estimate, based on interdependent RCMs, is approximately double that of the uncertainty generated by the GCMs.

The uncertainty estimate based on the RCMs has several weaknesses. First of all the ensemble is small, specially since the models are not independent from each other. That would tend to yield a small uncertainty. On the other hand the bulk of the uncertainty is due to two different clusters of solutions. The coupled models generally behave differently that the uncoupled ones. That increases the RCM uncertainty in a somewhat artificial way. Sea level in the Baltic Sea can be tuned to some extent (Meier et al., 2004; Gräwe and Burchard, 2012) and should eventually lead to a smaller uncertainty. Sea level is probably

the one variable in Baltic Sea or North Sea and Baltic Sea models that can be used for forecast, even without assimilation of data, at least direct assimilation into the ocean model. The atmospheric forcing is the other crucial ingredient and that is usually derived from the weather forecast. That is the regular procedure in the agencies concerned with sea level forecast around the North Sea and Baltic Sea.

If the RCM uncertainty was determined from the coupled models only without the coarse resolution version the uncertainty

was an order of magnitude smaller. Similar for the ocean-only models. There the uncertainty would be reduced by a factor of 2, except for stations Stockholm and Ringhals. Overall, the present, crude estimate of RCM uncertainty gives an upper bound of what can be expected from an analysis of a true multi-model ensemble. This effort should be tackled in the near future to understand better the uncertainty in ESLs inherent in the choice of the RCM. It would be important to assess the RCM uncertainty based on an ensemble of RCMs that have been driven with different GCMs. That would sample the full matrix and

would potentially uncover GCM/RCM combinations that yield ESLs outside of our basic estimate that uses one RCM only.

Another task concerns the reduction of the uncertainty that stems from the GCMs. In our ensemble we have used five different CMIP5 GCMs that span the parameter space. The addition of well behaved, but independent GCMs into the ensemble of regional projections would be valuable to generate more robust estimates and presumably a smaller uncertainty. In our ensemble the GCM uncertainty of 20 to 50 cm is larger at half of the stations than the confidence limits related to the estimation

of the 100-year return levels. Observationally based estimates of return levels are known to produce outliers on the Swedish





west coast (Fredriksson et al., 2016). It is not clear whether these are among the 5% that are bound to be outside the 95% confidence limits or whether the length of the time series or details of the algorithm must be improved. Arns et al. (2013); Vousdoukas et al. (2016); Wahl et al. (2017) have shown that the choice of the algorithm with which return levels are estimated can have a substantial impact on the result. On the other hand Lang and Mikolajewicz (2019) have shown for the German Bight
that 100-year return levels based on observations significantly underestimate the range of possible outcomes since they do not sample properly natural variability.

For planning and management purposes it is important to consider the spread of possible solutions along with the mean or median estimates. For ESLs along the Swedish coast we see potential for the reduction in uncertainty from both improvements on the RCMs and the representation of the climate by the GCMs. We have considered only one estimator for the 100-year return
levels, but Södling and Nerheim (2017) has shown that different approaches yield a range of results, also along the Swedish coast. These uncertainties should be taken into account as well in future investigations.

Recently, Meier et al. (2019) assessed different sources of uncertainty in projections of biogeochemical cycles in the Baltic Sea. Some of the uncertainties may be reduced by developing better modeling strategies, boundary and forcing data. Other uncertainties are related to unknown future nutrient input and greenhouse gas emissions. They stress the importance of regular
information on current knowledge which includes the uncertainty in model results that stem from different sources. In the context of high end climate change scenarios Capela Lourenço et al. (2018) did not find climate change uncertainty as being perceived as a barrier in the implementation of climate adaption. Uncertainty estimates are also planned to be included in management tools such as Symphony within the ClimeMarine project.

## 7 Conclusions

In this study we have analyzed ESLs in a regional sea level ensemble for the Baltic Sea. The ensemble uses one RCM forced with different GCMs. This allows to assess the uncertainty of 100-year return levels introduced by large scale circulation patterns represented by the GCMs. This uncertainty is one to two times the confidence limits of the observational GEV estimates. The observational confidence limits express the uncertainty in 100-year return levels from the use of short timeseries. Another source of uncertainty lies in the use of a specific RCM. We have estimated an upper bound for this uncertainty to be double the
size of the GCM uncertainty. With the analysis of sensitivity studies, processes and shortcomings have been identified that will allow model development to reduce this uncertainty below the GCM uncertainty.

The main findings of this study may be summarized as follows:

– The ensemble mean 100-year return levels range from 90 cm in the central Baltic Sea to 140 cm in the Bothnian Bay and southwestern Baltic Sea. These estimates are within within O(10 cm) and within the 95% confidence limits of
the observational estimates, except for the stations Gothenburg and Smögen. The uncertainty for 100-year return levels amounts to 20 to 50 cm.



- – The GCM uncertainty for the 99.9 percentile is largest where the ESLs are largest: In the Bothnian Bay, Gulf of Finland, Gulf of Riga and in the southwestern Baltic Sea. Along the Swedish coast the largest uncertainty is on the south and west coast of Sweden.

- – The GCM uncertainty of the ensemble mean 100-year return levels is the same order of magnitude as the 95% confidence limits from the GEV estimates with the blockmaxima method.

- – The bias in ESLs at stations Gothenburg and Smögen needs to be reduced. Sensitivity studies have shown that high frequency variability should be included at the open boundaries of the regional ocean model. Model development should also aim to reduce the RCM uncertainty.

*Code and data availability.* Data and software used for the analyses in this study can be made available from the authors upon request. The
10 code for the ocean model that is used in RCA4-NEMO is available at https://dx.doi.org/10.5281/zenodo.2643477.

*Author contributions.* The concept of the study was jointly developed by CD, MG, LA and HA. CD did the model runs, analysis, visualization and manuscript writing. MG contributed with literature analysis and methodology. All authors, CD, MG, LA and HA reviewed and edited the original draft.

*Competing interests.* The authors declare that they have no conflict of interest.

*Acknowledgements.* This study was funded by the Swedish Civil Contingencies Agency (MSB) through the project "HazardSupport: Risk-based decision support for adaptation to future natural hazards". The work has received funding from the project "Future flooding risks at the Swedish Coast: Extreme situations in present and future climate", Ref. No. P02/12 by Länsförsäkringars Forskningsfond. Support was provided from the project ClimeMarine founded by the Swedish Research Council Formas within the framework of the National Research Programme for Climate (grant no 2017-01949) and the Swedish Space Board within the project "Assimilating SLA and SST in an operational
ocean forecasting model for the North Sea and Baltic Sea using satellite observations and different methodologies" (grant no. 172/13).

The coupled model runs with RCA4-NEMO have been conducted on the Linux clusters Krypton, Bi and Triolith, all operated by the NSC [http://www.nsc.liu.se/]. Resources on the Linux cluster Triolith have been made available by the grant SNIC 002/12-25 "Regional climate modeling for the North Sea and Baltic Sea regions". This part of the simulations were performed on resources provided by the Swedish National Infrastructure for Computing (SNIC) at the National Supercomputer Centre in Sweden (NSC).



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
