# Peer review of "Extreme Sea Levels in the Baltic Sea under Climate Change Scenarios. Part 1: Model Validation and Sensitivity"

_Ocean Science, 2019_

## Referee Comment (RC1) · Anonymous Referee #1 · 22 Jul 2019

**General comments**

This paper describes results from a large number of climate simulations for the Balitic Sea. The writing standard is good, however I found it rather difficult to keep track of which set of tests were being discussed at a particular time due to the large number of different experiments that are being presented.

Additionally there are a number of references to events and datasets which are not properly introduced or explained for an unfamiliar reader.

[Figure]

**Specific comments**

Page 1 line 20 - What is "Backafloden"? This should be either removed, or explained.

Page 4 line 31 - Which storm surge model is being referred to here?

Page 7 Table 2 and related text - what is "ORAS4"?

Page 9 line 14 - What is the "WISKI network"?

Figures

Figure 1: The text labels are quite difficult to read here. It would be better if they were moved to be over the plain land rather than over the busy contours. This is often not easy to do, but alternatively they could be plotted with a semi-transparent background which may help. I am also not clear which experiments were used to calculate the ensemble mean—is it just the ones from Table 1?

Figures 2–8: These figures are all very similar to each other and it is not easy to tell at a glance what is being included in each one — it would be helpful to have an overview of what sensitivity it is showing (for example "Sensitivity tests using different RCMs", "Sensitivity tests using different ocean boundary conditions" etc). Additionally it would also be much clearer if the station names where included as titles on each subplot rather than only mentioned in the caption text, particulary since they are not included in the captions on Figures 4 and 5 so we have to keep going back to Figure 3 to see which station is which.

Is there a reason that Figures 3–6 show only 3 stations but the others show 9?

Figure 8: The black line (observation 99.9

**Technical corrections**

- Page 4 line 5: emphasize -> emphasis

- Pagew 23 line 17: differently that -> differently to/from/than (I am not sure which is most grammatically correct, I would probably use *to*)

---

## Referee Comment (RC2) · Anonymous Referee #2 · 29 Jul 2019

General comments

This study aims to validate the simulations of regional coupled circulation model by testing its extreme sea level outputs for the Baltic Sea region. For the downscaling of global model and of reanalysis outcomes, regional coupled model RCA4-NEMO is used. Results indicate that regional coupled model estimates on extreme sea levels are representing the observations on the stations, in general, well.

I think this paper focuses on very crucial topic for Baltic Sea, since the mechanisms driving mean sea level variability and extreme sea level variability are not well established for different regions of the Baltic Sea. Overall, the paper is well designed and

addresses a key topic of the Baltic Sea.

However, I think, there are some issues needed to be clarified in terms of content and writing style of the paper. In that sense, I tried to contribute to increase the quality of the paper by pointing some lines which can be found below. Hopefully it will be useful for authors in order to make the paper more reader friendly and clarified. (I think paper needs to be proofread by an external person. Given the case that there is a research gap on modelling MSL and ESL, this paper will be read by different authors in near future. Therefore, it would be very efficient to make this paper reader friendly)

First of all, the novelty of paper is not well written. Could authors please clearly write down why is this study important? (in Abstract and Introduction)

Secondly, the paper is a little bit hard to read. I have an impression that paper is written much quicker than it should be. Therefore, text is not distilled enough to easily follow the line of paper's story. For instance, first sentence of the abstract is already confusing. It basically claims that regional climate change scenarios are validated in the recent past. One may ask: how can you validate scenarios by comparing them with the recent past of extreme sea levels? I would try to formulate the first sentence in a way to avoid this complexity. It may be formulated like: We analyzed a regional climate model with respect to variability of recent extreme sea levels in the Baltic Sea. (if this sentence is true, it is much easier to read).

OR

'We investigated the variability of mean and extreme sea levels under different climate scenarios by using a regional coupled model for the Baltic Sea.' (Since the paper indeed also analyzes mean sea level variability)

Technical corrections

P1L3 (Page1Line23): Sentence:'…have been downscaled'. Is it dynamically or statistically downscaled? Please put it in front of the word 'downscaled'.

P1L4: Sentence 'Validation of 100-year….' Is also complicated to read. Do authors mean: 'Along the Swedish coast, simulations indicate a significant coherency of the model comparing to the estimations based on ESL observations, except for the sea level stations on the west coast, in terms of 100-year return periods of ESLs'?

P1L5: Sentence reads: 'The ensemble mean 100-year return levels turns out to be the best estimator with biases less than 10 cm.' I did not understand this sentence. Because up-to-now, this study only analyzes 100-year return levels. Or there are other return level periods that are analyzed in this study? If yes, which return periods?

P1L6: Sentence starts with: 'The ensemble spread..'. This sentence is redundant. It should be removed from the abstract.

P1L10: Please update the sentence starting in this line in this way: 'Some regions like Skagerrak, Gulf of Finland…'.

P1L14: which observational records? Tide gauges?

Authors use sometimes term 'sea level' in the text. Please make it clear whether you mention about 'mean sea level' or 'extreme sea level'.

I did not understand the sentence in P1L23: 'That has lead to harbours falling dry with economic impact on local societies.'. Do authors mean 'This relative fall of mean sea level rise along the harbours caused economic damages for local societies.'? Order of sentences in the paragraph is also confusing. Because authors first mention about increasing risk of being flooded, then say that indeed relative sea level is falling. It is hard to understand the logic behind it. Connections between paragraphs are also a bit weak. First paragraph can be put just before the paragraph starts with Analyses of ESLs by Weisse et al. (P3L1).

What does 100-year return level mean? I could not see any explanation about it. This is a concept which should be briefly explained in this paper.

P2L7 Please revise 'has coordinated to an ensemble' to 'has coordinated to an ensem-

ble mean'

P2L9 again add 'mean' to next to the word 'ensemble'

P2L13 Sentence starts at this line does not mention about the region. Where will be the magnitude of GIA and GMSL effects same in terms of relative sea level rise?

Can authors briefly explain the methods that they applied in this study at the end of introduction? (before P4L3).

P4L3: First two sentences of this paragraph should be placed earlier in the introduction section where the authors explain the novelty of this study.

P4L10: Which paper? Please put the reference. And please also write the principle conclusion of that paper, if it is needed to understand the scope of this paper.

P5L12 please cite the paper at the end of the sentence.

P8L11: Why Landsort is chosen, not Stockholm?

P8L18: Authors mention that modeled and observed sea surface for the period 1970-1999 is compared in Fig 1. You have observations only along the coasts. But interpretation of comparison covers the inner side of the Baltic Sea. Please explain which kind of assumption you did. Also show where is Skagerrak Where is Kattegat Where is Bothnian Bay? Please put them on the figure.

P15L1: What is a,b,c in Figure 5?

P16L2: Sentence: 'In the Baltic sea the impact on ESLs is negligible'. What is the subject of this sentence? I did not understand the sentence. This sentence is also a good sample representing the common mistakes in terms of writing style through whole paper. Writing part needs more attention than authors gave for this paper.

P23L7; It is mentioned that SLR,tides. Storm surges and wind waves increase the 100-year return levels more than SLR alone over the German Bight. I understand that

authors would like to show the importance of including high frequency variations in sea level, but German Bight is not in the Baltic Sea. For example, as far as I know, tides do not play an important role in driving sea level variability in the Baltic Sea. In that sense, please try to cite a study which has analysed high frequency sea level in the Baltic Sea. Or say why German Bight is representative for the Baltic Sea.

P23L25. Again the same issue. The sentence 'Similar for the ocean-only models'. Where is the subject of the sentence? I understand what authors mean, but they did not write exactly what they would like to say.

P23L35. I think sentence should start with 'Observation based' not 'Observationally based.' In Discussion section, it is mentioned that a second part of the study discusses the sea level projections. I could not understand what is the point of putting this information here. Because it is not further discussed and also study is not properly cited.

---

## Author Comment (AC1) · 9 Aug 2019

Answers to referee #1

Thank you for your comments and suggestions. We think it helped to improve the manuscript.

Page 1 line 20 - What is "Backafloden"? This should be either removed, or explained.

- Backafloden is the Swedish name for the storm surge in November 1872 that flooded a number of cities in the southwestern Baltic Sea with record sea levels. We have described the event in a sentence and avoided the name Backafloden.

[Figure]

Page 4 line 31 - Which storm surge model is being referred to here?

- The name of the storm surge model is NOAMOD. It is used at the SMHI for generating high frequency variability in the northern North Sea for the sea level forecast models. We have added a sentence to the text about NOAMOD. Unfortunately, there is no reference that could be cited.

Page 7 Table 2 and related text - what is "ORAS4"?

- ORAS4 is the previous Ocean Reanalysis System evaluated in e.g. Balmaseda et al., 2013. We have now included the reference in the text and in the caption, together with the explanation of the acronym.

Page 9 line 14 - What is the "WISKI network"?

- In this and most other cases we have replaced "WISKI database" or "WISKI network" with "tide gauge network". The name WISKI appears now only together with the reference to SMHI's open data.

Figure 1: The text labels are quite difficult to read here. ...

- We have moved the labels for the stations into free spaces on the map. The caption now mentions that the ensemble mean consists of the historical periods of the scenarios in Table 1.

- Figure 1b now contains labels for the different basins of the Baltic Sea instead of the station names. This is to accommodate a suggestion of reviewer #2.

Figures 2–8: These figures are all very similar to each other ...

- We agree. We have added a short characterization of the figures at the beginning of the captions. Figures 2 to 8 include now the station names.

Is there a reason that Figures 3–6 show only 3 stations but the others show 9?

- The three stations were meant as characteristic examples for the Bothnian Bay, Baltic

[Figure]

Proper and the west coast. The only reason to reduce those figures to three panels was to save space and text. We show now nine stations for all figures. The descriptions of the figures in the text have been expanded to include some sensitivities of the new figures.

Figure 8: The black line (observation 99.9

- We have adapted the caption of Figure 8.

References:

- Balmaseda, M. A., Mogensen, K., and Weaver, A. T.: Evaluation of the ECMWF ocean reanalysis system ORAS4, Q J Roy Meteor Soc, 139, 1132-1161, https://doi.org/doi:10.1002/qj.2063, 2013

---

## Author Comment (AC2) · 9 Aug 2019

Answers to referee #2

Thank you for your detailed comments and suggestions. It helped to improve different aspects of the manuscript including readability.

General comments

However, I think, there are some issues needed to be clarified in terms of content and writing style of the paper. ...

- We have reworked the manuscript to streamline it and improve the consistency and

style. In many places the reviewer's suggestions have helped to eliminate redundant formulations or to add necessary explanations.

First of all, the novelty of paper is not well written. ...

- We have added a paragraph towards the end of the introduction that specifies more clearly what this study contributes to the ESL research in the Baltic Sea.

- The beginning of the abstract now also highlights the contribution to the uncertainty discussion.

Secondly, the paper is a little bit hard to read. ...

- This is a good point. The beginning of the article should be inviting to read on. We have simplified the first sentence and made other changes to the abstract that are meant to improve the readability.

Technical corrections

P1L3 (Page1Line23): Sentence:'. . .have been downscaled'. ...

- We have introduced 'dynamically' downscaled in the sentence.

P1L4: Sentence 'Validation of 100-year. . ..' Is also complicated to read. ...

- Agreed. We have reformulated the sentence.

P1L5: Sentence reads: 'The ensemble mean 100-year return levels turns out to be the best estimator with biases less than 10 cm.' ...

- The idea here was to mention that the ensemble mean of an ensemble of model runs shows a better agreement with observations than a single model run, even the ERA40 hindcast. We have simplified the sentence.

P1L6: Sentence starts with: 'The ensemble spread..'. This sentence is redundant. It should be removed from the abstract.

- We think this is among the important findings of our study. The 5% to 95% range of the ensemble solutions (the likely range in IPCC parlance) includes the estimates based on observations. This gives us some confidence that the solutions in the ensemble cover what has been observed. This will be important in cases where there are no observations. We have combined this finding with the summary of the GCM uncertainty at the end of the abstract.

P1L10: Please update the sentence starting in this line in this way: 'Some regions like Skagerrak, Gulf of Finland. . .'.

- We have changed the text according to the reviewer's recommendation.

P1L14: which observational records? Tide gauges?

- We have clarified in the text that we mean tide gauge records.

I did not understand the sentence in P1L23:

- Yes, the text was mentioning relative sea level rise and fall due to two different processes. We have now removed the example on sea level fall due to GIA to improve the readability. The estimates for GIA appear now after the introduction on GMSL rise.

- We prefer to leave the first paragraph at the beginning of the introduction. It explains why we want to know about SLR and ESL.

What does 100-year return level mean? ...

- Yes, it is true, we did not explain the return level concept. We have added a short paragraph to Section 3.2 (Extreme Sea Levels) where the return levels are introduced.

P2L7 Please revise 'has coordinated to an ensemble' to 'has coordinated to an ensemble mean.'

- We think that the formulation as it is describes better that the CMIP5 project (Taylor et al., 2012) has coordinated an ensemble of model runs with GCMs. From this ensemble

we have downscaled five individual ensemble members (GCMs) with a regional climate model and calculated return periods that we want to discuss.

P2L9 again add 'mean' to next to the word 'ensemble'

- We would like to mention in the text, that many aspects of the ensemble of GCMs are discussed in IPCC's AR5. This includes also the ensemble spread (the uncertainty) not only the ensemble mean.

P2L13 Sentence starts at this line does not mention about the region. ...

- The effect of GIA and GMSL on Baltic Sea level depends on the region in the Baltic Sea and on the climate change scenario. We have tried to describe this in the sentences that follow.

Can authors briefly explain the methods that they applied in this study at the end of introduction? (before P4L3).

- The main method that was used to generate the model estimates for the ESLs was the dynamical downscaling of a number of GCMs. This is part of the paragraph (previously) starting at P5L9 in Section 2. We left out the details of how the dynamical downscaling and the coupling between atmosphere and ocean was done. This has been published in e.g. Wang et al., 2015, Dieterich et al., 2019.

- We first followed the suggestion and summarized our approach with the statistical methods in the paragraph on novelty in the introduction (see also next answer). That would have required to introduce the return period concept in the introduction. We have decided against it and prefer to explain the return period concept where we show the first return levels for different return periods (Tab. 7, Sect. 3.2). To avoid repetition we have now added a brief outline of our approach at the beginning of Section Extreme Sea Levels, not in the introduction.

P4L3: First two sentences of this paragraph should be placed earlier in the introduction section where the authors explain the novelty of this study.

- We have moved the two sentences further up in the introduction and combined them with the description of the novelty of this study.

P4L10: Which paper? Please put the reference. And please also write the principle conclusion of that paper, if it is needed to understand the scope of this paper.

- We have added a short explanation at the end of the introduction how this paper is related to the companion paper. We also added the reference. The second part is not needed to understand the results and discussion in this paper. The second part gives an extra motivation for this study.

P5L12 please cite the paper at the end of the sentence.

- We have added the reference for the paper, which is in preparation.

P8L11: Why Landsort is chosen, not Stockholm?

- We chose Landsort because it is usually taken as the reference station in the middle of the Baltic Sea, near the nodal line, that represents the amount of water (or the mean sea level) in the Baltic Sea, e.g. Lisitzin, 1974., Meier et al., 2004, Mohrholz et al., 2015. It is also the station with the least amount of spread in ESLs among different ensemble members, cf. Fig. 2 and 7. The tide gauge station in Stockholm is close to the city center, where it can be affected by freshwater from lake Mälaren (Samuelsson and Stigebrandt, 1996) and where the model resolution is too coarse to properly resolve the Stockholm archipelago.

P8L18: Authors mention that modeled and observed sea surface for the period 1970-1999 is compared in Fig 1. ...

- We validate the model results for MSL along the coast, where observations from tide gauge stations are available. The model results show a reasonable agreement with observations. We implicitly assume that MSLs along the coast are tightly connected to MSLs in adjacent regions in the open Baltic Sea. Those are mainly determined by the shallow water equations including the atmospheric and riverine forcing. These are

properly represented in our model, although there is room for improvement. To avoid confusion we have changed the wording in the paragraph and call the observations mean sea level, not mean sea surface. We have not used a mean sea surface based on observations.

- We have updated Figure 1b) to show the names of the different gulfs and basins in the Baltic Sea that are mentioned in the text.

P15L1: What is a,b,c in Figure 5?

- We have now included station labels in Figures 2 to 8 to make it easier to identify individual stations. This was a suggestion of reviewer #1.

P16L2: Sentence: 'In the Baltic sea the impact on ESLs is negligible'. ...

- We have eliminated this sentence.

- We have tried improve the text in many places to make it more readable.

P23L7; It is mentioned that SLR,tides. Storm surges and wind waves increase ...

- The idea was to show an example where the interaction of waves and sea level has a large effect. That's why we chose to cite Arns et al., 2017. We agree that the interaction with tides is not significant in the Baltic Sea. We have replaced the sentence with examples from the Baltic Sea, that have been described in Weisse and Weidemann, 2017, Viitak et al., 2016, Wisniewski and Wolski, 2011 and Averkiev and Klevanny, 2007.

P23L25. Again the same issue. The sentence 'Similar for the ocean-only models'. ...

- We have changed the text to make the statement clearer.

P23L35. I think sentence should start with 'Observation based' not 'Observationally based.' ...

- We have changed the wording in this sentence to 'Observation based'.

- The reason for mentioning the second part of the study was to provide a clue for the interested reader to the scenario part of the model ensemble in connection with the uncertainty discussion. We have now eliminated the sentence.

References:

- Taylor, K. E., Stouffer, R. J., and Meehl, G. A.: An Overview of CMI P5 and the Experiment Design, B Am Meteorol Soc, 93, 485–498, https://doi.org/10.1175/BAMS-D-11-00094.1, 2012.

- Wang, S., Dieterich, C., Döscher, R., Höglund, A., Hordoir, R., Meier, H. E. M., Samuelsson, P., and Schimanke, S.: Development and evaluation of a new regional coupled atmosphere-ocean model in the North Sea and Baltic Sea, Tellus A, 67, https://doi.org/10.3402/tellusa.v67.24284, 2015.

- Dieterich, C., Wang, S., Schimanke, S., Gröger, M., Klein, B., Hordoir, R., Samuelsson, P., Liu, Y., Axell, L., Höglund, A., and Meier, H. E. M.: Surface Heat Budget over the North Sea in Climate Change Simulations, Atmosphere, 10, https://doi.org/10.3390/atmos10050272, 2019.

- Lisitzin, E.: Sea-Level Changes, Elsevier Oceanography Series vol. 8, Amsterdam, 1974.

- Meier, H. E. M., Broman, B., and Kjellström, E.: Simulated sea level in past and future climates of the Baltic Sea, Clim Res, 27, 59–75, https://doi.org/10.3354/cr02705904, 2004.

- Mohrholz, V., Naumann, M., Nausch, G., Krüger, S., and Gräwe, U.: Fresh oxygen for the Baltic Sea - An exceptional saline inflow after a decade of stagnation, J Marine Syst, 148, 152–166, https://doi.org/10.1016/j.jmarsys.2015.03.005, 2015.

- Samuelsson, M. and Stigebrandt, A.: Main characteristics of the long-term sea level variability in the Baltic sea, Tellus A, 48, 672–683, https://doi.org/10.1034/j.1600-0870.1996.t01-4-00006.x, 1996.

- Viitak, M., Maljutenko, I., Alari, V., Suursaar, Ü., Rikka, S., and Lagemaa, P.: The impact of surface currents and sea level on the wave field evolution during St. Jude storm in the eastern Baltic Sea, Oceanologia, 58, 176–186, https://doi.org/https://doi.org/10.1016/j.oceano.2016.01.004, 2016.

- Averkiev, A. S. and Klevanny, K. A.: Determining cyclone trajectories and velocities leading to extreme sea level rises 10 in the gulf of Finland, Russian Meteorology and Hydrology, 32, 514-519, https://doi.org/10.3103/S1068373907080067, 2007.

- Weisse, R. and Weidemann, H.: Baltic Sea extreme sea levels 1948-2011: Contributions from atmospheric forcing, Procedia IUTAM, 25, 65-69, https://doi.org/10.1016/j.piutam.2017.09.010, 2017.

- Wisniewski, B. and Wolski, T.: Physical aspects of extreme storm surges and falls on the Polish coast, Oceanologia, 53, 373–390, https://doi.org/10.5697/oc.53-1-TI.373, 2011.

---

## Author Comment (AC3) · 9 Aug 2019

The comment was uploaded in the form of a supplement:
https://www.ocean-sci-discuss.net/os-2019-65/os-2019-65-AC3-supplement.pdf